# The Divider Assay is a high-throughput pipeline for aggression analysis in *Drosophila*

Budhaditya Chowdhury[1], Meng Wang [1], Joshua P. Gnerer[1] & Herman A. Dierick [1,2✉]

Aggression is a complex social behavior that remains poorly understood. *Drosophila* has become a powerful model system to study the underlying biology of aggression but lack of high throughput screening and analysis continues to be a barrier for comprehensive mutant and circuit discovery. Here we developed the Divider Assay, a simplified experimental procedure to make aggression analysis in *Drosophila* fast and accurate. In contrast to existing methods, we can analyze aggression over long time intervals and in complete darkness. While aggression is reduced in the dark, flies are capable of intense fighting without seeing their opponent. Twenty-four-hour behavioral analysis showed a peak in fighting during the middle of the day, a drastic drop at night, followed by re-engagement with a further increase in aggression in anticipation of the next day. Our pipeline is easy to implement and will facilitate high throughput screening for mechanistic dissection of aggression.

[1] Department of Molecular and Human Genetics, Baylor College of Medicine, One Baylor Plaza, Houston, TX 77030, USA. [2] Department of Neuroscience, Baylor College of Medicine, One Baylor Plaza, Houston, TX 77030, USA. ✉email: dierick@bcm.edu

Animal model systems have been crucial to understand behavior and elucidate its basic mechanisms. With its easy and cheap husbandry, rich behavioral repertoire[1], powerful genetics and gene editing[2–4], and an ever more sophisticated ability to regulate neural circuitry during behavior[5–7], *Drosophila melanogaster* continues to be a valuable model organism to study behavior. Key to its success has been the use of screens to identify genes and circuits that control behavior[8–24]. High-throughput screens depend on assays that are fast and accurate, but such assays are harder to design for complex behaviors such as aggression. Aggressive male flies display three unambiguous aggressive behaviors: wing threat, lunging, and boxing[25,26], of which lunging is the predominant one[27]. Since the first detailed description of these behaviors in flies[25], a number of methods have been developed to quantify the behavior[26,28–34]. The time to run an experiment and analyze the data determine how fast aggression can be measured and thus how many strains can be tested. Automated video analysis[31,33,35] has made data analysis faster, but has not altered the time it takes to run an experiment, therefore impeding high-throughput analysis.

Here we report a novel pipeline to quantify aggression that maximizes throughput without compromising accuracy and reliability. Recently eclosed males are introduced into a shallow chamber with square arenas covered by a glass plate and separated by dividers to keep the flies isolated. The assembled chamber rests on a layer of clear food that provides sustenance for the flies during isolation. After several days of isolation, dividers are gently removed and flies are videotaped. Using an existing machine-learning paradigm (JAABA)[36], we developed a classifier to precisely quantify lunging behavior, even during the high intensity boxing bouts, when both flies engage in rapid mutual lunging. We validated our classifier against independent manual observers and an existing, mostly rule-based quantification system (CADABRA)[33]. We directly compared the Divider Assay with two established methods and examined different assay parameters used in various existing assays. Our pipeline is faster, more accurate, and easier to implement than existing methods. In addition, the setup allows to record behavior in complete darkness making long-term round the clock recordings of aggression possible. We discovered that flies are able to fight without seeing their opponent and have a 24-h rhythmic variation in aggression similar to courtship and mating behavior[37,38]. Our pipeline provides a standardized method to measure aggression and will dramatically improve screening for mutant and circuit discovery.

## Results

### Divider Assay: 3D-printed chamber and classifier for precise quantification of aggression.
Reliably measuring aggression in flies requires five sequential steps (Fig. 1a). The experimenter must: (1) collect recently eclosed flies; (2) isolate the flies individually on food to increase future aggression and prevent starvation; (3) aspirate previously isolated flies in pairs into an arena of a chamber that can be videotaped; (4) record the interactions of loaded pairs in the chamber for 20 min; and (5) analyze the recordings to identify aggression events per pair for further quantification and statistical analysis. Lunging is the most common behavioral feature of fighting males[27] and most existing aggression assays measure lunging behavior[23,28,32,33] or generate a score depending on lunging behavior[34,39]. Of these five steps, quantifying lunges is the most time-consuming one, and can require hours of video analysis per genotype. Automated video analysis[31,33,35] has greatly reduced the amount of time spent to measure aggression. However, the first three steps take at least as long as the recording step and this has hampered implementation of aggression assays in high-throughput screens.

To overcome the challenge of time-consuming behavioral experimentation, we developed the Divider Assay. We designed a 3D-printed behavior chamber that lets the experimenter execute the first three steps enumerated above simultaneously in less than 5 min (diagrammed in Fig. 1a, assembly shown in Supplementary Video 1 and Supplementary Data 2). Our fighting chambers hold 12 fighting pairs in separate square arenas. The shallow square shape of the arenas eliminates the need for fluon coating (further reducing preparation time) to increase the flies' interaction time. Opaque, thin, stiff dividers that slide through each square arena maintain future interacting pairs in an isolated state, making them more prone to aggression[32,40]. After spending 5 days in social isolation, the dividers are gently removed and flies are recorded for 20 min. We processed the behavioral recordings in three different ways to measure lunge numbers (Fig. 1b): (1) we manually counted lunge numbers derived from 60 pairs with varying fighting intensities through slow-motion video analysis (this is the established gold standard against which any software method is compared); (2) we developed our own quantification pipeline taking advantage of existing fly-tracking software (FlyTracker)[35] and machine-learning paradigm (JAABA)[36] with an added filtering step (see "Methods"); and (3) we used CADABRA, which is an existing automated software system that depends in part on rule-based quantification[33] (Fig. 1b).

To compare the performance of our newly developed classifier, we separated the fighting flies in four categories depending on the manual lunge number measurements in the 20-min videos: (1) 0–20 lunges, very low or non-fighting flies; (2) 21–100 lunges, moderately fighting pairs; (3) 101–300 lunges, high fighting pairs; and (4) more than 300 lunges, which are very aggressive flies that lunge every few seconds throughout the entire recording period. In all categories, there was no significant difference in performance between our classifier (green boxplots) and the gold standard manual (yellow boxplots) lunge count (Fig. 1c, Kruskal–Wallis ANOVA with Dunn's test, 0–20, $p = 0.60$; 21–100, $p = 0.41$; 101–300, $p = 0.32$; >300, $p = 0.79$, $n = 15$ per group, Supplementary Video 2 illustrates raw lunging and boxing data and classifier performance in both contexts). In contrast, CADABRA (red boxplots) showed a trend to overscore very low fighting pairs (Fig. 1c, Kruskal–Wallis ANNOVA with Dunn's test, 0–20, $p = 0.17$, $n = 14$) and significantly underscored moderate to very high fighting pairs (Fig. 1c, Kruskal–Wallis ANNOVA with Dunn's test; 21–100: $p = 0.040$; 101–300: $p < 0.0001$; >300, $p < 0.0001$, $n = 15$ per group). Nevertheless, we detected strong highly significant correlations between manual scoring and automated quantification for our classifier ($R^2 = 0.98$, $p < 0.0001$, $n = 60$) as well as CADABRA ($R^2 = 0.87$, $p < 0.0001$, $n = 59$) (Fig. 1d) showing that correlation analysis is not sufficient to evaluate assay performance. The significant error in lunge quantification of CADABRA for the different fighting intensities can best be visualized with estimation plots (Gardner–Altman plots), which shows the effect size of the scoring error[41] (see "Methods" and Supplementary Fig. 1).

Finally, we manually curated a low and a high fighting pair frame-by-frame to evaluate how many lunges were misclassified by our classifier. For both types of flies, the false-positive and false-negative rates were very similar (~5% and ~8%, respectively), showing that the classifier performs with high accuracy and sensitivity (Fig. 1e). Taken together these data show that the Divider Assay precisely quantifies lunge numbers over a broad range of fighting intensities.

### Aggression decreases with increased space.
We next wanted to carefully compare the Divider Assay with existing assays and started by evaluating spatial parameters as these vary between the

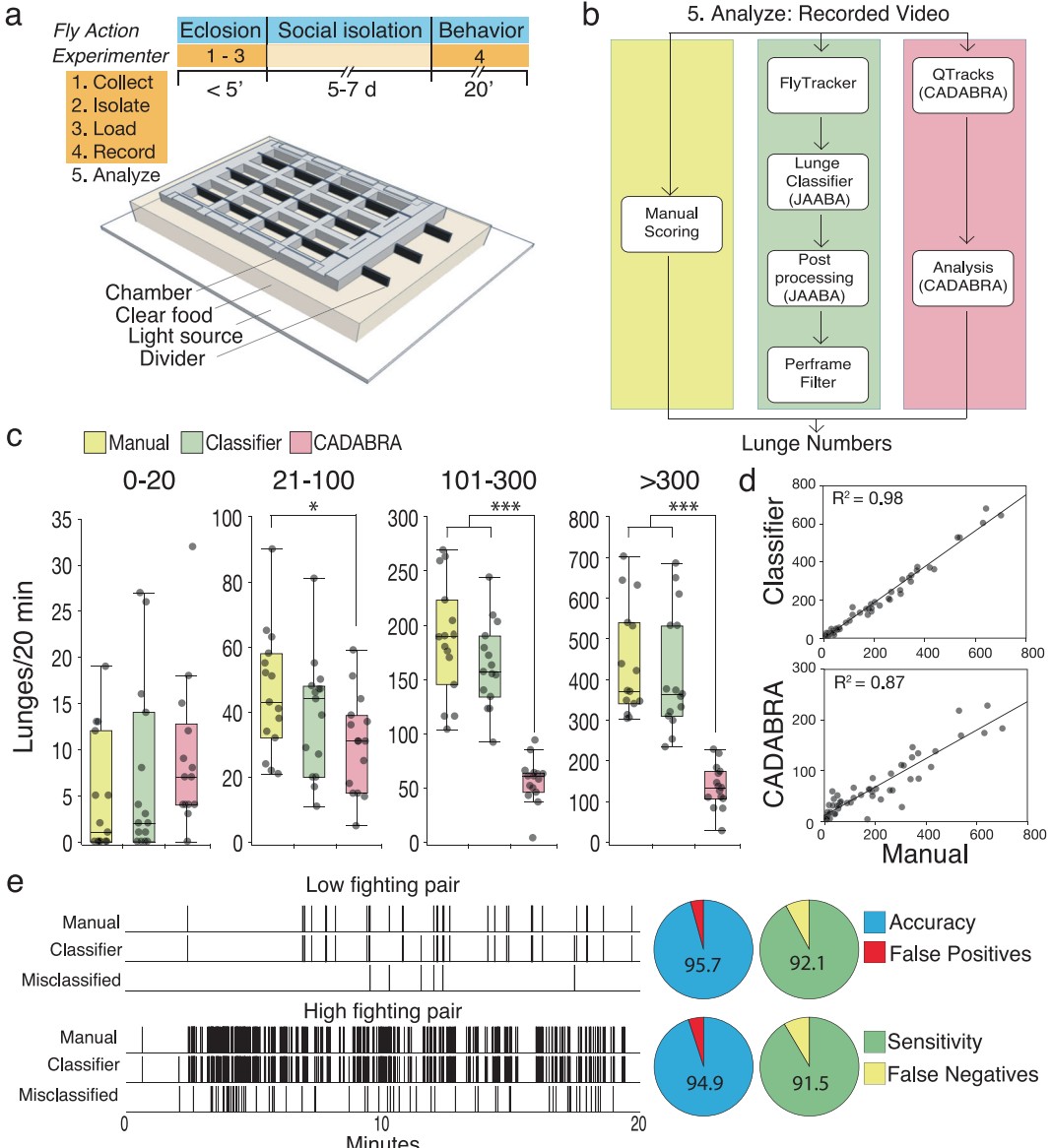

**Fig. 1 Divider Assay for precise quantification of aggression over a broad range of fighting intensities. a** Schematic of the Divider Assay setup and experimental time line. Collecting, isolating, and loading flies are done in one step under 5 min. Flies are then isolated on clear food until they are videotaped. **b** Recorded video analysis can be done manually (yellow), with a newly developed JAABA-based classifier to precisely score lunges (green), or with existing CADABRA software (red). **c** Sixty pairs of flies with lunges ranging from 0 to 700 were analyzed manually, with a new JAABA-based classifier, and CADABRA. For this analysis, four groups of fighting intensities were chosen: 0–20, 21–100, 101–300, and >300 lunges. The JAABA-based classifier performed close to the gold standard with no significant differences in any group (Kruskal–Wallis ANOVA with Dunn's test and Bonferroni correction 0–20, $p = 1.0$; 21–100, $p = 0.55$; 101–300, $p = 0.75$; >300, $p = 1.0$, $n = 15$ per group). CADABRA tended to overscore low fighting flies, but significantly underscored high fighting pairs (Kruskal–Wallis ANOVA with Dunn's test and Bonferroni correction, 0–20: $p = 0.08$, $n = 14$; 21–100: $p = 0.03$; 101–300: $p < 0.0001$; >300: $p < 0.0001$, $n = 15$ per group) (see also Supplementary Fig. 1, for Gardner–Altman estimation plots). **d** Regression analysis between manual scoring and classifier ($R^2 = 0.98$, $p < 0.0001$, $n = 60$) or CADABRA ($R^2 = 0.87$, $p < 0.0001$, $n = 59$) are both highly significant. **e** Misclassified lunges with the JAABA-based classifier occur at low frequency in both low and high fighting pairs (false positives ~5% and false negatives ~8%). Boxplots show the median, first and third quartiles as boxes, with whiskers representing the 5 and 95% intervals.

different methods currently used in the field. Our standard chamber is 4.5 mm high to afford fighting flies enough space to stand up on their hindlegs, which is common during aggressive interactions. The chamber contains an array of 4 × 3 square arenas of ~1.3 cm wide (1.69 cm$^2$).

We first examined the effect of changing height and surface area and measured the effect on aggression of both a low aggression control strain (Canton S, CS) and a hyper-aggressive strain (see "Methods") to make sure our assay works well over a broad range of fighting intensities. We varied height from

3.5 mm, the height used in the FlyBowl assay[42], to 11 mm, the height used in the previously developed Arena Assay[39], and statistically compared all the results to our standard 4.5 mm chamber. Even though 3.5 mm is just above the height of a wild-type fly with raised legs standing on its hindlegs, aggression did not significantly change in this condition in either strain (Fig. 2a, b, nonparametric Steel method with the standard 4.5 mm chamber height as the control, $4.5^{CS}$ vs. $3.5^{CS}$, $p = 0.34$; and $4.5^{Aggr}$ vs. $3.5^{Aggr}$, $p = 1.0$, $n = 24$). Adding an additional fly length in the 7.5 mm chamber or more than two fly lengths to

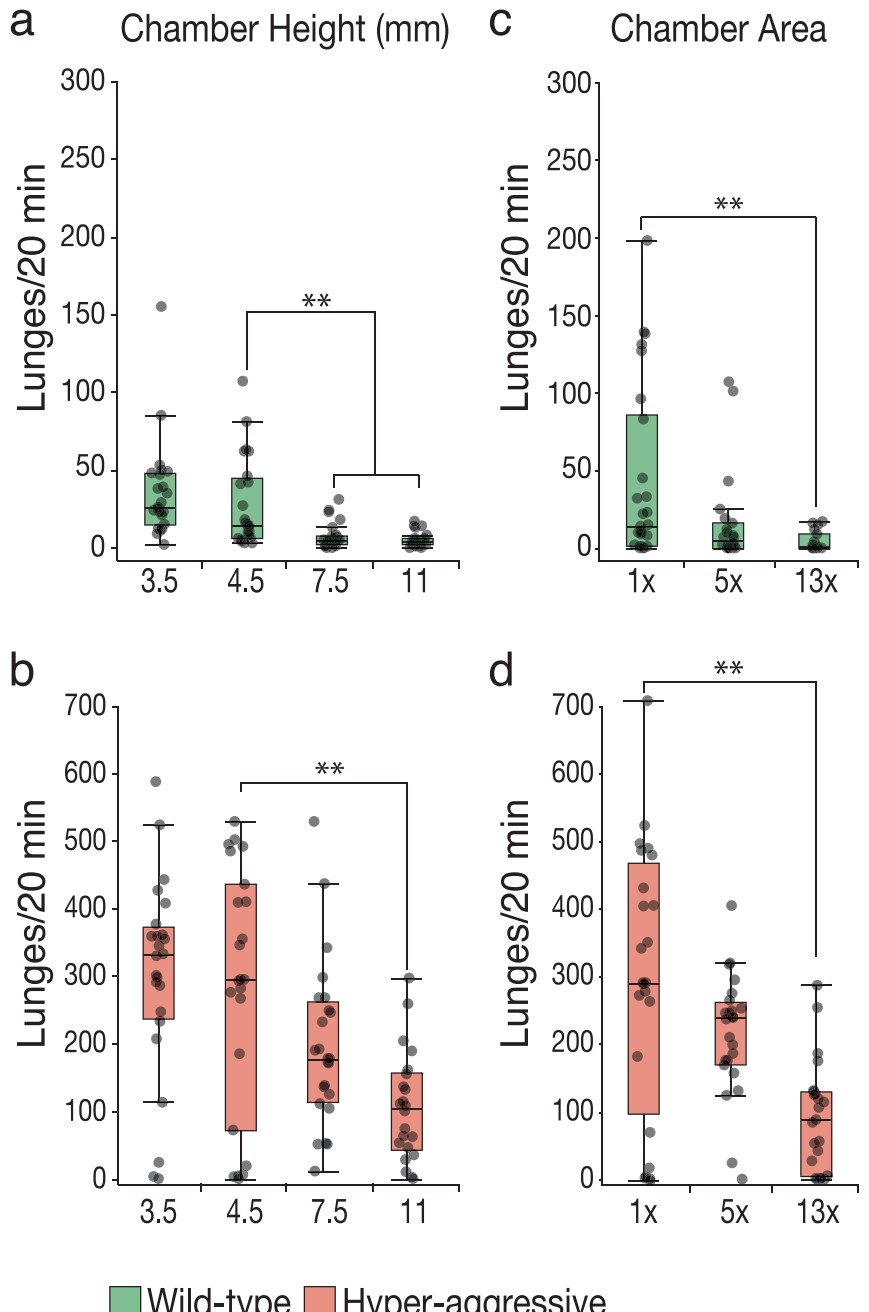

**Fig. 2 Increased space decreases aggression. a** Lunge numbers decrease with increasing height of the square arenas for low fighting control strain. There is no difference between 4.5 and 3.5 mm, which is approximately the height of a fly raised on its hindlegs (nonparametric Steel method with 4.5 mm as control height, $p = 0.34$). As the height of the arena increases to 7.5 and 11 mm lunge numbers statistically significantly decrease (nonparametric Steel method compared to 4.5 mm as a control, $7.5^{CS}$, $p = 0.001$; $11^{CS}$, $p < 0.001$). **b** For hyper-aggressive flies, the decrease only became statistically significant at 11 mm (nonparametric Steel method compared to 4.5 mm height as control, $3.5^{Aggr}$, $p = 1$; $7.5^{Aggr}$, $p = 0.11$; $11^{Aggr}$, $p = 0.005$, $n = 22–24$ per group). **c**, **d** Increasing the surface area of the arenas ~5 to 13-fold by doubling or tripling the width of the arena also decreased aggression. In both low and high aggression strains, the difference becomes statistically significant when the arena width is tripled (nonparametric Steel method with 1× surface area as the control, $1×^{CS}$ vs. $5×^{CS}$, $p = 0.07$; vs. $13×^{CS}$ $p = 0.002$; $1×^{Aggr}$ vs. $5×^{Aggr}$, $p = 0.07$; vs. $13×^{Aggr}$ $p = 0.001$, $n = 17–24$ per group). Boxplots show the median, first and third quartiles as boxes, with whiskers representing the 5 and 95% intervals.

11 mm as in the Arena Assay however decreased aggression. The decrease with one additional fly length was significant in wild-type CS low aggression flies, but not in hyper-aggressive flies, while the 11-mm height significantly decreased aggression in both low and high aggression strains (Fig. 2a, b, nonparametric Steel method with control, $4.5^{CS}$ vs. $7.5^{CS}$, $p = 0.001$; vs. $11^{CS}$, $p < 0.0001$; $4.5^{Aggr}$ vs. $7.5^{Aggr}$, $p = 0.11$; vs. $11^{Aggr}$ $p = 0.005$, $n = 22–24$ per group). We next increased surface area by doubling or

tripling the width of the arena, which increases surface area ~5 times (9 cm$^2$) and ~13 times (22 cm$^2$), respectively (Supplementary Data 3). A fivefold increase in arena surface decreased aggression although the decrease did not reach statistical significance. A 13-fold increase in surface area led to a statistically significant drop in aggression in both strains (Fig. 2c, d, nonparametric Steel method with 1× standard surface area as the control, $1×^{CS}$ vs. $5×^{CS}$, $p = 0.07$; vs. $13×^{CS}$ $p = 0.002$; $1×^{Aggr}$

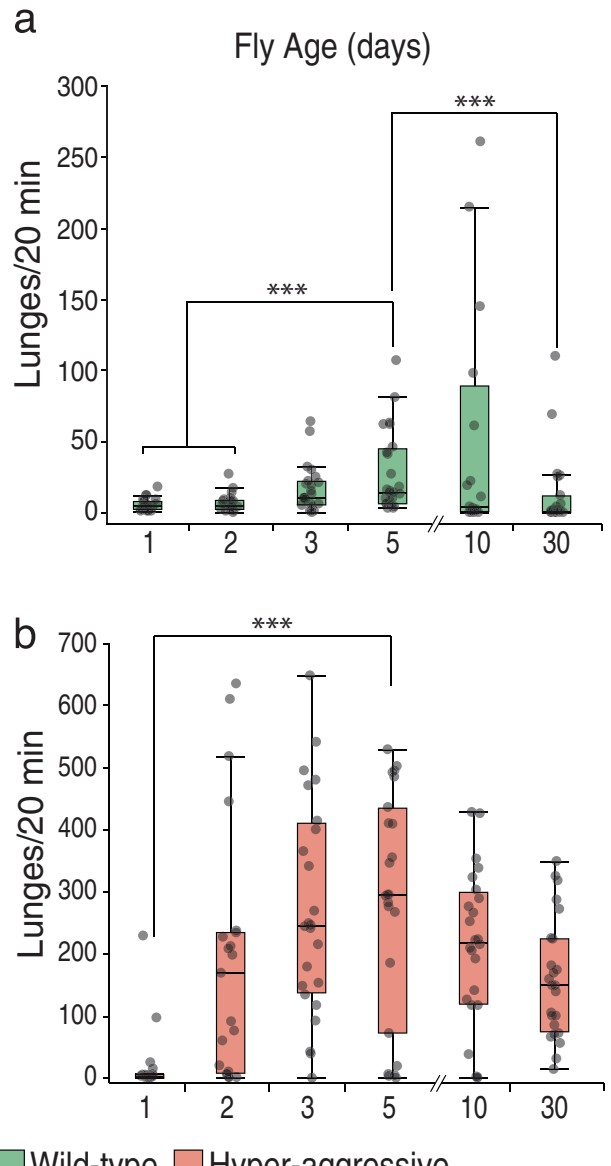

**Fig. 3 Old flies remain aggressive. a** Control flies peak by 5 days and decline after but some flies still fight by 30 days of age. There is no significant difference in fighting between 5- or 10-day-old flies, but all other ages are significantly lower than 5-day-old flies (nonparametric Steel method with the 5-day-old as a control, $5d^{CS}$ vs. $1d^{CS}$, $p = 0.0002$; $5d^{CS}$ vs. $2d^{CS}$, $p = 0.0011$; $5d^{CS}$ vs. $3d^{CS}$, $p = 0.57$; $5d^{CS}$ vs. $10d^{CS}$, $p = 0.50$; $5d^{CS}$ vs. $30d^{CS}$, $p = 0.0017$, $n = 19$–24 per group). **b** Compared to control flies hyper-aggressive flies already fight a lot by 2 days of age, but further significantly increase in aggression by 5 days. Even at 30 days, these flies still lunge more than 150 times in 20 min (nonparametric Steel method with the 5-day-old as a control, $5d^{Aggr}$ vs. $1d^{Aggr}$, $p < 0.0001$; $5d^{Aggr}$ vs. $2d^{Aggr}$, $p = 0.23$; $5d^{Aggr}$ vs. $3d^{Aggr}$, $p = 0.92$; $5d^{Aggr}$ vs. $10d^{Aggr}$, $p = 0.19$; $5d^{Aggr}$ vs. $30d^{Aggr}$, $p = 0.06$, $n = 21$–24 per group). Boxplots show the median, first and third quartiles as boxes, with whiskers representing the 5 and 95% intervals.

vs. $5\times^{Aggr}$, $p = 0.07$; vs. $13\times^{Aggr}$ $p = 0.001$, $n = 17$–24 per group). Taken together these results show that even a hyper-aggressive strain fights less when there is more space available in the arena.

**Old flies are capable of high levels of aggression.** The age of the flies that are tested for aggression also varies across different assay systems, so we next examined the effect of age on aggression levels in the Divider Assay. Typically in most assay systems, 5- to 7-day-old flies are used because flies below 3–4 days of age fight very little[43] and flies older than 1 week are generally considered too old as most behaviors decline rapidly with fly age[44,45]. We examined aggression in flies at six different ages: 1, 2, 3, 5, 10, and 30 days post eclosion. Flies were loaded into the divided arenas on the day of eclosion, except for the 10- and 30-day-old flies, which were loaded 3 days before they were videotaped. Aggression peaked in 5-day-old flies for both strains and declined after although the decline at 10 days was not significant in either strain (Fig. 3a, nonparametric Steel method with the 5-day-old as the control, $5d^{CS}$ vs. $1d^{CS}$, $p = 0.0002$; $5d^{CS}$ vs. $2d^{CS}$, $p = 0.0011$; $5d^{CS}$ vs. $3d^{CS}$, $p = 0.57$; $5d^{CS}$ vs. $10d^{CS}$, $p = 0.50$; $5d^{CS}$ vs. $30d^{CS}$, $p = 0.0017$, $n = 19$–24 per group). Even at 2 days of age, hyper-aggressive flies already fight in excess of 10 times more than the wild-type controls at their peak. While the levels of aggression in this strain were lower than at their 5-day-old peak (Fig. 3b, nonparametric Steel method with the 5-day-old as the control, $5d^{Aggr}$ vs. $1d^{Aggr}$, $p < 0.0001$; $5d^{Aggr}$ vs. $2d^{Aggr}$, $p = 0.23$; $5d^{Aggr}$ vs. $3d^{Aggr}$, $p = 0.92$; $5d^{Aggr}$ vs. $10d^{Aggr}$, $p = 0.19$; $5d^{Aggr}$ vs. $30d^{Aggr}$, $p = 0.06$, $n = 21$–24 per group), the 2-day-old flies were isolated for a shorter period of time. To evaluate the extent to which the 5-day-old phenotype is due to age or isolation, we repeated the experiment and compared aggression in 2-day-old flies that were isolated for 2 days to 5-day-old flies that were either isolated for 2 days or for 5 days. The median lunge number in the 5-day-old flies isolated for 2 days was nearly identical to that in the 2-day-old flies and lower than the 5-day-old flies isolated for longer (Supplementary Fig. 2, Kruskal–Wallis ANOVA, $p = 0.29$, $n = 24$ per group). These results suggest that the effect of isolation is more important than age, at least at this time point. Remarkably, even at 30 days of age, the hyper-aggressive flies lunged more than once every 4 s. We previously showed that these flies fight even in group housed conditions and accumulate wing damage[24]. To minimize this effect, we group housed the flies before loading in groups of five. Together these results show that aggression does not show a dramatic decline even as flies reach very old age.

**Comparison of the Divider Assay to several existing assay systems.** We next compared the Divider Assay to two established methodologies, the Colosseum Assay[28] and the Arena Assay[29,39]. Several parameters differ between these assays: size of the chamber, age of the flies, presence of food, isolation duration, loading of the flies, method of lunge analysis, and duration of the experiment (Fig. 4b). We quantified both lunge number and latency to fight. In the Colosseum and Arena Assays, analysis was done manually while in the Divider Assay, we used our newly developed classifier. All assay systems showed similar results although in the Colosseum Assay the differences between the strains were less pronounced likely because of the larger space in the chamber consistent with our findings when we increased space in the Divider Assay (Fig. 4a, Colosseum$_{Lunge}$, $p = 0.0001$; Arena Assay$_{Lunge}$, $p < 0.0001$; Divider Assay$_{Lunge}$, $p < 0.0001$; Colosseum$_{Latency}$, $p = 0.005$; Arena Assay$_{Latency}$, $p < 0.0001$; Divider Assay$_{Latency}$, $p < 0.0001$, Wilcoxon rank-sum test). In addition, latency to lunge for the wild-type strain is significantly shorter in the Colosseum Assay compared to the Arena Assay (Wilcoxon rank-sum test, $p = 0.002$) likely because of the presence of a food source in the former, which is lacking in the latter.

**Assay duration affects aggression.** We next wanted to evaluate the effect of assay duration as some investigators have used assays of 10 min and some even as short as 2 min or have focused their

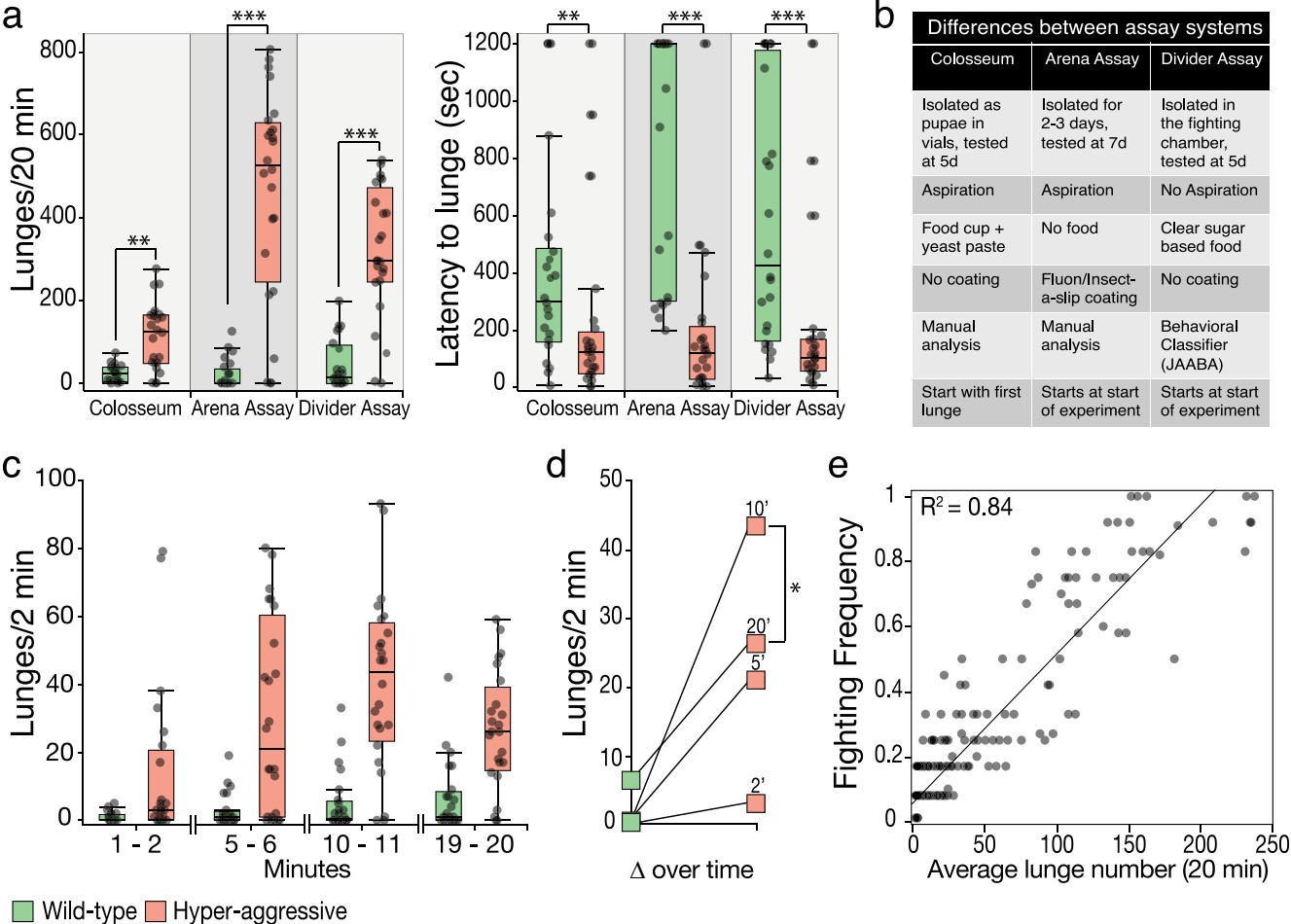

**Fig. 4 Comparison of the Divider Assay with other aggression assays. a** Lunge numbers (left plot) and latencies to lunge (right plot) in two established assays, the Colosseum assay and the Arena Assay, show similar differences between control and hyper-aggressive strains as in the Divider Assay. In all assays the differences are statistically significant (Wilcoxon rank-sum test, $n = 22$–24 per group). **b** Overview of the parameter differences in the three assay systems: age, fly handling, presence or absence of food, method of analysis, and duration of the experiment are different in the three assays. The Divider Assay is easiest to run because flies are collected, isolated, and loaded in a single step and lunges are analyzed in an automated manner. **c** Analysis of lunges in 2 min time bins shows significant differences in lunge numbers between the low and high aggression strains (Wilcoxon rank-sum test, 1–2 min, $p = 0.001$; 5–6 min, $p = 0.0009$; 10–11 min, $p < 0.0001$; 19–20 min, $p < 0.0001$, $n = 24$). **d** Plot of the median differences between the low (green squares) and high (red squares) aggression strains The largest difference between the two strains occurs at the 2 min bin at 10 min of videotaping. Median lunge numbers decrease significantly by 20 min in the hyper-aggressive strain (Wilcoxon rank-sum test, Aggr$^{10'}$ vs. Aggr$^{20'}$, $p = 0.03$, $n = 24$). In the low aggression strain median lunge number keep increasing over time and reach the highest levels in the last 2 min time bin (Wilcoxon rank-sum test, CS$^{2'}$ vs. CS$^{20'}$, $p = 0.02'$, Wilcoxon rank-sum test, $n = 24$). **e** Fighting frequencies, the percentage of pairs that show clear dominance between both flies, correlates very significantly with lunge numbers as previously shown ($R^2 = 0.84$, $p < 0.0001$, 10–12 fighting pairs per data point, $n = 213$). Boxplots show the median, first and third quartiles as boxes, with whiskers representing the 5 and 95% intervals.

analysis on a 2–3 min window in a 10-min assay, to make the analysis easier and faster[30,32,34]. We quantified lunges in a sliding 2-min window to look at fight dynamics over time and to evaluate whether we could still pick up the strong difference in aggression between our wild-type control and hyper-aggressive strains. Even in the earliest 2 min window, we found a statistically significant difference in the lunge number between both strains, but the difference between the strains increased significantly over time (Fig. 4c, 1–2 min, $p = 0.001$; 5–6 min, $p = 0.0009$; 10–11 min, $p < 0.0001$; 19–20 min, $p < 0.0001$; Wilcoxon rank-sum test, $n = 24$). The largest difference in median lunge number occurred at 10 min (Fig. 4d) and aggression decreased significantly by 20 min in the hyper-aggressive strain (Fig. 4c, d, Aggr$^{10'}$ vs. Aggr$^{20'}$, $p = 0.03$, Wilcoxon rank-sum test, $n = 24$). In the wild-type low aggression strain, lunge numbers increased over time and reached the highest level at 20 min compared to the first 2 min window (Fig. 4c, d, CS$^{2'}$ vs. CS$^{20'}$, $p = 0.02$, Wilcoxon rank-sum

test, $n = 24$). Together these results suggest that decreasing the assay duration below 10 min is not advisable and that looking at the first two min would likely lead to many false-negative differences between strains.

Another lunge derived metric shown to strongly correlate with lunge numbers is fighting frequency[24]. Analysis from our Divider Assay confirms this result demonstrating that frequency largely depends on lunges as the main aggression parameter (Fig. 4e, $R^2 = 0.84$, $p < 0.0001$, 10–12 fighting pairs per data point, $n = 213$).

**Mean aggression score (MAS) does not measure aggression.** The shortest assay described to date to analyze aggression is a group assay in which all "aggressive" interactions between 4 and 8 males are evaluated[30]. The males are first starved for 1.5 h in an empty food vial and are next provided with a small amount of yeast paste by changing the lid of the vial. The investigators then manually count the interactions of males competing for food and

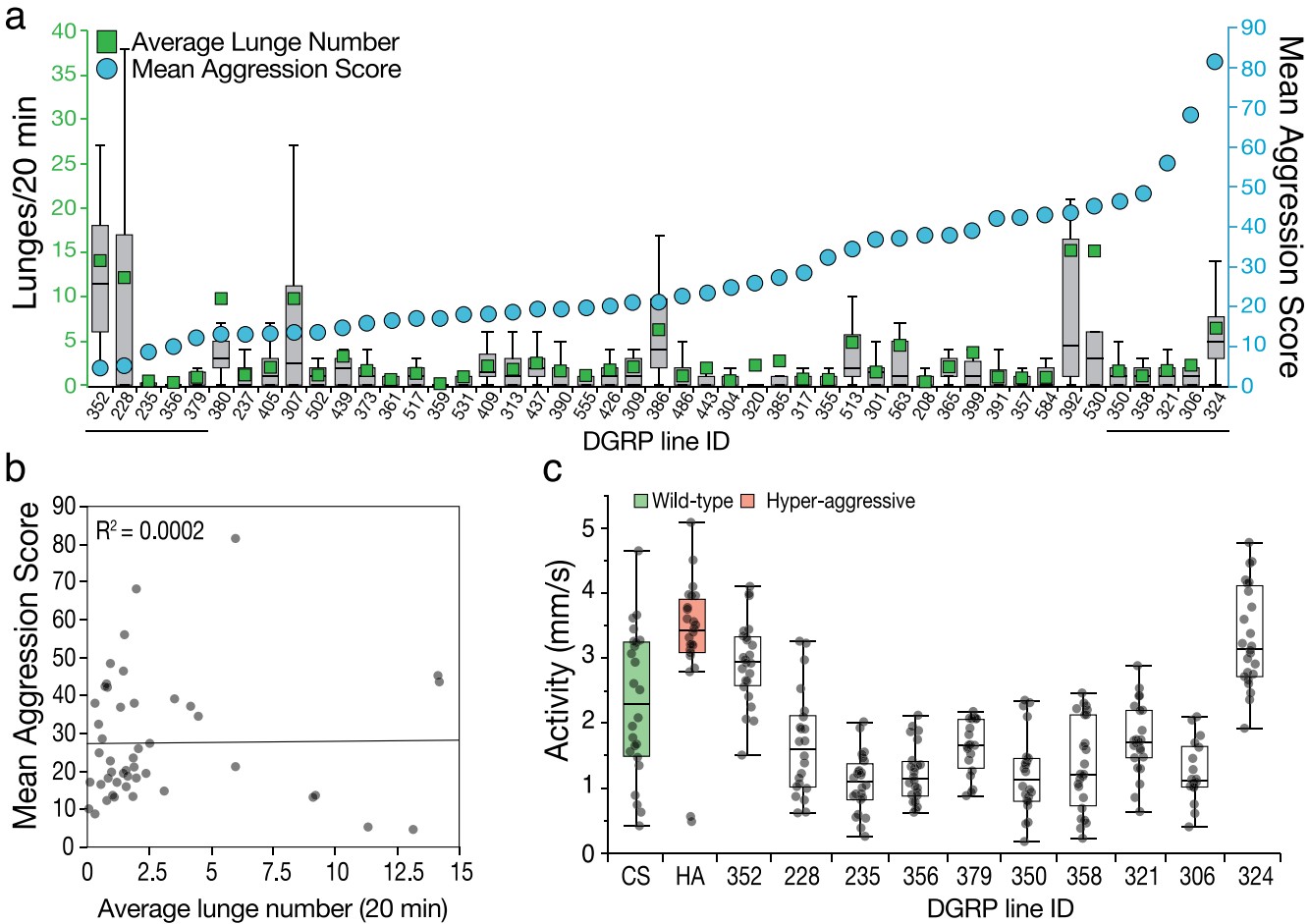

**Fig. 5 Mean Aggression Score (MAS) does not measure aggression. a** Median (line in the box) and average lunge (green square) numbers of 47 DGRP strains that were previously analyzed for MAS (blue circles). None of the strains have lunge numbers above 15 in 20 min interval (very low or no aggression in the range of CS). MAS varies 20-fold from ~4 in the lowest strain to ~80 in the highest strain. **b** Regression analysis shows that both parameters—lunge number and MAS—are not correlated ($R^2 = 0.0002$, $n = 12$–24). **c** Average locomotion data over 20 min of the five DGRP strains with lowest MAS and five DGRP strains with the highest MAS compared to the low (green horizontal bar) aggression and hyper-aggressive (red horizontal bar) strains. Most of the DGRP strains have significantly lower locomotor activity than CS (nonparametric Steel method with CS as the control, 352, $p = 0.35$; 228, $p = 0.33$; 235, $p = 0.0007$; 356, $p = 0.0054$; 379, $p = 0.30$; 350, $p = 0.0077$; 358, $p = 0.027$; 321, $p = 0.42$; 306, $p = 0.018$; 324, $p = 0.044$; Aggr, $p = 0.02$). Boxplots show the median, first and third quartiles as boxes, with whiskers representing the 5 and 95% intervals.

express them as an MAS. They used this method to profile the Drosophila Genetic Reference Panel (DGRP) developed in their lab[46] to identify loci implicated in aggression[47,48].

To evaluate whether any of these lines have elevated lunge numbers, we profiled a subset of the DGRP panel and performed lunge analysis. We selected 50 of the ~200 strains to represent the full spectrum of MAS scores from very low to very high to best capture the correlation with lunge analysis. We obtained lunge numbers from 47 strains because some of the strains grew very poorly. Surprisingly, we found that MAS scores did not overlap at all with average or median lunge number (Fig. 5a). None of the lines showed lunge numbers that exceeded the aggression scores of our low aggression CS strain despite showing an MAS range of more than 20-fold between the lowest and highest lines included in the set we analyzed[48]. Regression analysis of the MAS scores and lunge number averages showed a near zero value (Fig. 5b, $R^2 = 0.0002$, $p = 0.92$, $n = 47$). It is highly unlikely that our method fails to detect aggression in these strains because we successfully scored aggression in other previously published strains[49] with increased aggression (Supplementary Fig. 3) and we reliably detect low and medium lunge numbers (Fig. 1c). We also looked at locomotor activity for a selected subset (5 lowest, and 5 highest ranked lines

for MAS, which also spans their range of lunge numbers) of DGRP lines and found that compared to our control and hyper-aggressive strains, most of the DGRP lines we tested moved significantly less than our low and high aggression strains (Fig. 5c, nonparametric Steel method with CS as the control, 352, $p = 0.35$; 228, $p = 0.33$; 235, $p = 0.0007$; 356, $p = 0.0054$; 379, $p = 0.30$; 350, $p = 0.0077$; 358, $p = 0.027$; 321, $p = 0.42$; 306, $p = 0.018$; 324, $p = 0.044$; Aggr, $p = 0.02$; Supplementary Fig. 4). Together these results show that the MAS does not capture the main behavioral feature associated with aggression in Drosophila.

**Aggression shows 24-h rhythmic variation.** In earlier experiments where we evaluated lunge number over different 2 min time windows during a 20-min experiment, we discovered that peak aggression in the hyper-aggressive strain occurs after 10 min and significantly decreases towards the end of the assay, although the levels remain very high (Fig. 4c, d). This suggested to us that habituation may develop between pairs that have established a dominance relationship. We wanted to explore this further by examining flies over a 24-h interval. Our new assay allows us to profile flies over extended time periods since a food source is available. We profiled 24 pairs of flies starting at 8 AM and

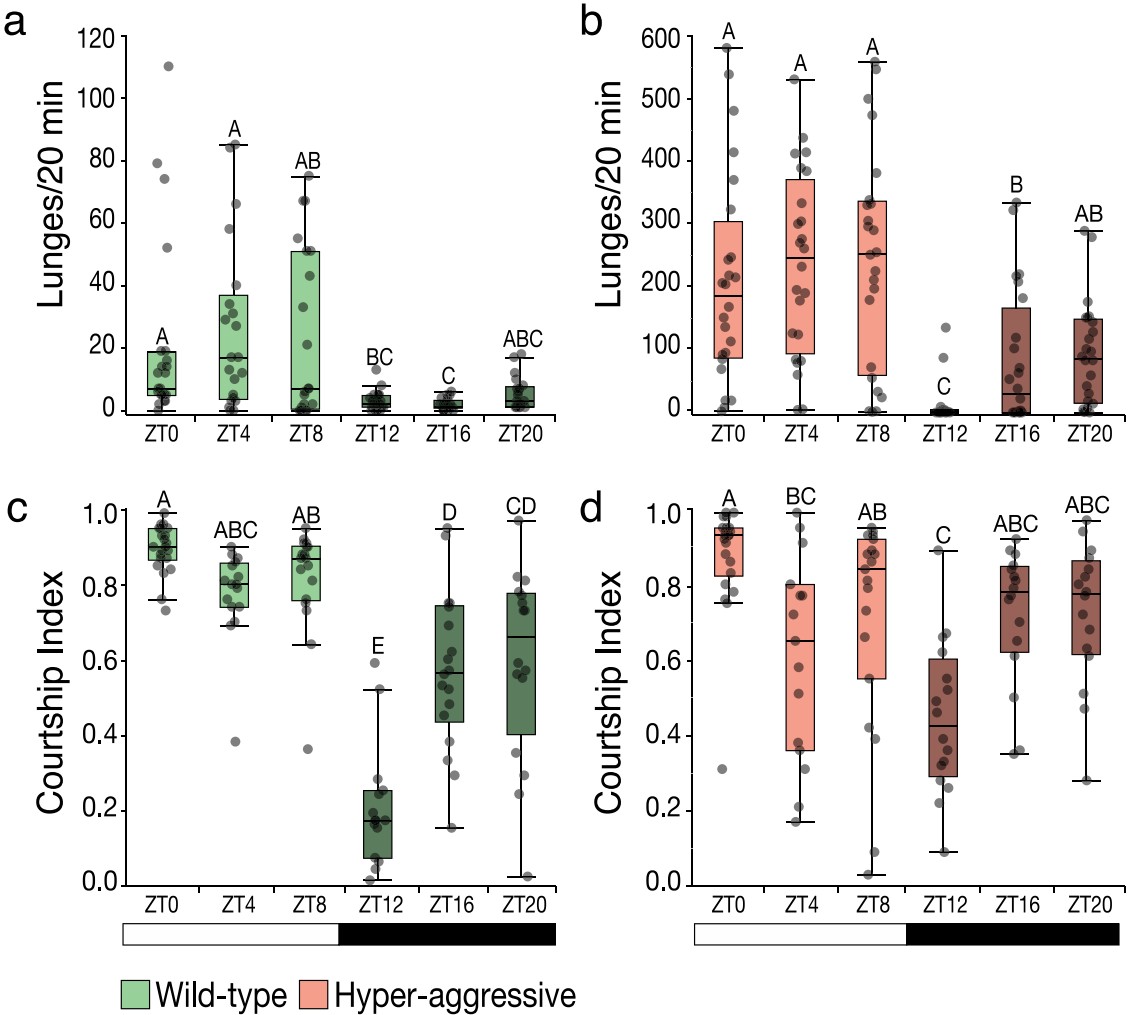

**Fig. 6 Aggression shows 24 h variation. a** Lunge number variation over the different times of the day. In the low aggression strain, lunge numbers are highest during the day with a trend to peak at ZT4. At night lunges decrease significantly and increase again in anticipation of the next day (Kruskal–Wallis ANOVA with Dunn's test and Bonferroni correction, significant differences are denoted with letters, ZT16 vs. ZT8, $p = 0.026$; ZT12 vs. ZT0, $p = 0.0018$; ZT12 vs. ZT4, $p = 0.0015$; ZT16 vs. ZT0, $p = 0.0002$; ZT16 vs. ZT4, $p = 0.0002$, $n = 20$–24 pairs per time point). **b** The lunge number variation follows a similar pattern but is more pronounced in the hyper-aggressive strain (Kruskal–Wallis ANOVA with Dunn's test and Bonferroni correction, ZT20 vs. ZT12, $p = 0.018$; ZT20 vs. ZT0, $p = 0.017$; ZT16 vs. ZT4, $p = 0.015$; ZT16 vs. ZT8, $p = 0.017$; ZT12 vs. ZT0, $p < 0.0001$; ZT12 vs. ZT4, $p < 0.0001$; ZT12 vs. ZT8, $p < 0.0001$, $n = 24$ pairs per time point). **c** The pattern of daily lunge number variation is similar to the variation in courtship index. In the low aggression strain, courtship is highest early in the day with a statistically significant decrease around noon, a steep drop at the beginning of the night, courtship then increases again in the middle of the night with a further although not significant increase in anticipation of the day (Kruskal–Wallis ANOVA with Dunn's test and Bonferroni correction, ZT16 vs. ZT8, $p = 0.026$; ZT20 vs. ZT0, $p = 0.0003$; ZT16 vs. ZT0, $p < 0.0001$; ZT12 vs. ZT4, $p = 0.0003$; ZT12 vs. ZT8, $p < 0.0001$; ZT12 vs. ZT0, $p < 0.0001$, $n = 15$–22 pairs per time point). **d** The same pattern can be observed in the hyper-aggressive strain with peak courtship in the early morning, a statistically significant decrease at noon, a steep decrease in the beginning of the night with recovery later in the night (Kruskal–Wallis ANOVA with Dunn's test and Bonferroni correction, ZT12 vs. ZT8, $p = 0.019$; ZT4 vs. ZT0, $p = 0.0047$; ZT12 vs. ZT0, $p < 0.0001$, $n = 15$–22 pairs per time point). Boxplots show the median, first and third quartiles as boxes, with whiskers representing the 5 and 95% intervals.

continually filmed them throughout the day and night until the next morning. As suggested by our 20 min experiment (Fig. 4d), over 24 h there is a strong decrease in aggression that falls to near zero levels by nightfall and continues to be low during the night (Supplementary Fig. 5a). Each time point is the combination of lunges in the three 20 min intervals per hour. Between the first 20 min of the first hour and the last 20 min of the second hour, the drop in lunge numbers is statistically significant (Supplementary Fig. 5b, Wilcoxon rank-sum test, $p = 0.04$). After the second hour median lunge numbers are approximately 10-fold lower than in the first 20 min interaction interval.

Because we found such a dramatic decrease in lunges over time, we wondered whether the time of day may also contribute to these low levels of aggression observed in the 24-h profile. We

therefore tested a new set of flies from both control and hyper-aggressive strains every 4 h throughout the day and night for a total of six time points. Since flies are loaded when the experiment is set up and dividers can be removed in the dark, these night-time recordings are now feasible whereas in other set-ups aspirating flies in the dark is difficult to impossible. As for the previous 24-h continuous lunge analysis experiments, flies were maintained at a 12 h/12 h light dark cycle under constant humidity. Both strains showed peak aggression levels at ZT4 with lower levels at ZT0 and ZT8 although the differences were not significantly different (Fig. 6a, b, Kruskal–Wallis ANOVA with Dunn's test and Bonferroni correction, $CS^{ZT12}$ vs. $CS^{ZT4}$, $p = 0.0015$; $Aggr^{ZT12}$ vs. $Aggr^{ZT4}$, $p < 0.0001$, $n = 24$ pairs per time point). Strikingly, at ZT12 aggression was near zero as in the

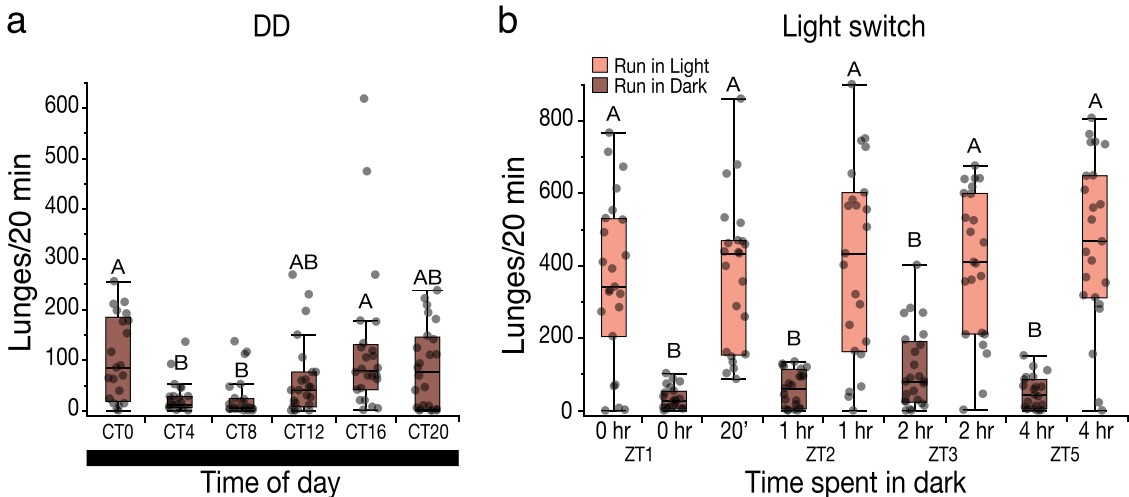

**Fig. 7 Aggression under free-running conditions compared to the effect lights "ON" and "OFF". a** Lunge number variations of the hyper-aggressive strain at different times of the day under free-running DD conditions (Kruskal–Wallis ANOVA with Dunn's test and Bonferroni correction, CT0 vs. CT8, $p = 0.0024$; CT0 vs. CT4, $p = 0.033$; CT16 vs. CT8, $p = 0.0009$; CT16 vs. CT4, $p = 0.0154$; CT20 vs. CT4, $p = 0.0983$; CT12 vs. CT4, $p = 0.5288$; CT20 vs. CT4, $p = 0.7187$, $n = 20$–$24$ pairs per time point). **b** Lunge numbers in flies that are run in light vs. dark at the beginning of the day significantly decrease strongly compared to flies run in light and flies kept in dark for 20 min and are then run in light (Kruskal–Wallis ANOVA with Dunn's test and Bonferroni correction, $ZT1^{0L}$ vs. $ZT1^{0D}$, $p < 0.0001$; $ZT1^{0L}$ vs. $ZT1^{20'L}$, $p = 1.000$; $ZT1^{0D}$ vs. $ZT1^{20'L}$, $p < 0.0001$, $n = 24$ pairs per time point). Increasing the dark treatment duration does not significantly increase lunge numbers of dark tested (Kruskal–Wallis ANOVA with Dunn's test and Bonferroni correction, all dark comparisons, $p = 1.000$, $n = 24$ pairs per time point). Dark treatment followed by testing the flies in light leads to fully recovered lunge numbers (Kruskal–Wallis ANOVA with Dunn's test and Bonferroni correction, all light comparisons, $p = 1.000$, $n = 23$–$24$ pairs per time point). Boxplots show the median, first and third quartiles as boxes, with whiskers representing the 5 and 95% intervals.

continuous experiment (Supplementary Fig. 5a), but increased again in the middle of the night at ZT16, particularly in the hyper-aggressive strain, and further increased at ZT20 in anticipation of the morning. A similar trend was also visible in the control low aggression strain.

Because similar 24-h rhythmicity has been observed in male courtship[37,50] and female mating behavior[38], we explored changes in courtship index (CI) measured across a 24-h period in our behavior chamber. For both wild type and hyper-aggressive lines, CI showed 24-h rhythmicity similar to what we observed for aggression, with the lowest levels observed at dusk (ZT12), with subsequent recovery later at night (Fig. 6c, d, Kruskal–Wallis ANOVA with Dunn's test and Bonferroni correction, $CS^{ZT16 \text{ vs. } ZT8}$, $p = 0.026$; $CS^{ZT20 \text{ vs. } ZT0}$, $p = 0.0003$; $CS^{ZT16 \text{ vs. } ZT0}$, $p < 0.0001$; $CS^{ZT12 \text{ vs. } ZT4}$, $p = 0.0003$; $CS^{ZT12 \text{ vs. } ZT8}$, $p < 0.0001$; $CS^{ZT12 \text{ vs. } ZT0}$, $p < 0.0001$, $Aggr^{ZT12 \text{ vs. } ZT8}$, $p = 0.019$; $Aggr^{ZT4 \text{ vs. } ZT0}$, $p = 0.0047$; $Aggr^{ZT12 \text{ vs. } ZT0}$, $p < 0.0001$, $n = 24$ pairs per time point). However, unlike aggression, there was a drop in CI at mid-day (ZT4), compared to ZT0 levels that reached statistical significance in the hyper-aggressive strain but not in the wild-type Canton S ($CS^{ZT0 \text{ vs. } ZT4}$ $p = 0.16$; $Aggr^{ZT0 \text{ vs. } ZT4}$ $p = 0.0047$). The latency for males to start courting did not show any such rhythmicity in 24 h for either strain (Supplementary Fig. 6c, d).

We next examined 24-h rhythmicity in mating frequency using our behavioral setup. We observed similar rhythmicity in mating frequency, which is considered a read-out for female choice. Both wild type and aggressive strains showed high mating frequency during the three light time points, a significant drop at dusk (ZT12), and recovery at the remaining two night-time points (Supplementary Fig. 6a, b, Kruskal–Wallis ANOVA with Dunn's post hoc test with Bonferroni correction, $CS^{ZT12 \text{ vs. } ZT4}$, $p = 0.0028$; $Aggr^{ZT12 \text{ vs. } ZT4}$, $p = 0.002$, $n = 7$ replicates of 6–7 pairs each). Together these results show that like courtship and mating

behavior, aggression follows a day/night variation with peak aggression mid-day and lowest levels at dusk.

**Free running rhythmic variation compared to the effect of lights ON and OFF.** We next examined whether the 24 h variations in aggression were maintained under free-running conditions. For these experiments, we maintained flies in constant darkness throughout the experiments and tested them every 4 h at the same relative time points as in the LD experiments. Circadian variation seen in LD conditions was still maintained, with the lowest levels of aggression shifted to earlier time points CT4 and CT8 when lunge numbers showed the lowest median (Fig. 7a, Kruskal–Wallis ANOVA with Dunn's test and Bonferroni correction, $Aggr^{CT0}$ vs. $Aggr^{CT4}$, $p = 0.033$; $Aggr^{CT0}$ vs. $Aggr^{CT8}$, $p < 0.0024$, $n = 20$–$24$ pairs per time point).

While aggression levels still showed variations under free-running conditions, overall levels were strongly reduced, indicating that light itself strongly affects aggression. We therefore examined the effect of turning lights off and back on and after increasing dark exposure. When lights are turned off in the morning, aggression levels plummet. Flies that were simultaneously put in the dark and kept there for 20 min and then tested with lights on, immediately rebounded to pre-dark treatment levels (Fig. 7b, Kruskal–Wallis ANOVA with Dunn's test and Bonferroni correction, $ZT1^{0L}$ vs. $ZT1^{0D}$, $p < 0.0001$; $ZT1^{0L}$ vs. $ZT1^{20'L}$, $p = 1.000$; $ZT1^{0D}$ vs. $ZT1^{20'L}$, $p < 0.0001$, $n = 24$ pairs per time point). Longer dark exposure increased lunge medians in dark run flies, but not significantly so in contrast to free-running dark tested flies. Similarly flies that were maintained longer and longer but run in light conditions all rebounded to very similar levels regardless of the duration of dark exposure (Fig. 7b, Kruskal–Wallis ANOVA with Dunn's test and Bonferroni correction, $ZT1^{0L}$ vs. $ZT1^{0D}$, $p < 0.0001$; $ZT1^{0L}$ vs. $ZT1^{20'L}$, $p = 1.000$; $ZT1^{0D}$ vs. $ZT1^{20'L}$, $p < 0.0001$, $n = 23$–$24$ pairs per time point).

## Discussion

We developed a novel pipeline for simple and reproducible high-throughput aggression analysis in *Drosophila*. We designed the Divider Assay using a cheap 3D-printed chamber with shallow, square arenas with central dividers so that flies can be loaded and separated as they eclose. After several days of isolation, the dividers are removed and 12 fly pairs are simultaneously filmed for 20 min. The chamber design makes the preparation time to set up a behavioral experiment easy and fast, cutting the time to run an experiment at least in half. Using the JAABA platform[36] we developed an accurate classifier that precisely scores lunges from the video files, is easy to run in currently available operating systems, and outperforms the previous automated video analysis software[33]. Lunging is the key fighting parameter between aggressive flies. Very aggressive flies also box, which represents reciprocal lunging, and is also measured precisely by our new classifier.

We used our platform to examine different assay parameters that have been used in aggression analysis in flies in different assay systems and to compare the performance of different assays on wild-type low aggression flies and a very high aggression strain derived from these control flies[29] (see also "Methods" for details). We show that our Divider Assay is as effective at scoring differences between the strains as several existing assays, but much easier to execute. Analysis of the data in 2-min intervals throughout the 20 min of video recording of fighting pairs shows that the difference between low and high strains is strongest after 10 min of interaction. One assay used in the literature measures MAS[30] in a 2-min time window between four to eight pairs of flies that were previously starved. The MAS was used to identify sequence variants in the DGRP collection that may explain the behavioral differences in these inbred strains[47,48]. We profiled a subset of these strains using our novel pipeline and show that lunge numbers in the profiled DGRP strains show no correlation with their MAS that correlates very well between the first analysis in 2009 and subsequent analysis in 2015. These findings show that MAS is not a reliable measure of aggression in flies and that the molecular findings from this previous work are unlikely to explain mechanisms of aggression.

Our 2-min interval analysis also showed that aggressive strains show a significant decrease in fighting in the second half of the 20 min of video recording, suggesting habituation between fighting flies. We further confirmed this by recording flies continuously for 24 h. After a few hours flies almost completely stop fighting although a small percentage of flies fight throughout the day and the night. We next also examined whether fighting is modulated by the time of day, as was recently shown in mice[51], and found that peak aggression occurs in the middle of the day, decreases drastically towards the end of the day and the beginning of the night, but then picks up again in anticipation of the next day. This day–night modulation follows a similar pattern as in courtship and mating behavior[37,38]. While turning lights off is likely the strongest contributor to the decrease in aggression in the night-time points, even under free-running conditions circadian modulation is maintained albeit slightly shifted to earlier time points. Future experiments will be needed to resolve the mechanistic underpinnings of the light switch effect.

*Drosophila* behavioral assays often go through phases of innovation and re-invention to make them more precise, easier to implement, and amenable for high-throughput screening. Such advances have promoted a refined understanding of behavior at molecular, neuronal, and circuit levels. In this study we developed an improved high-throughput pipeline to analyze aggressive behavior that is easy to implement across laboratories to expand the mechanistic dissection of this complex social behavior.

## Methods

**Fly stocks and husbandry.** The DGRP lines were obtained from the Bloomington Drosophila Stock Center. The following strains were tested: RAL 208 (BL#25174), 228 (BL#28157), 235 (BL#28275), 237, 301 (BL#25175), 304 (BL#25177), 306 (BL#37525), 307 (BL#25179), 309 (BL#28166), 313 (BL#28180), 317 (BL#28167), 320 (BL#29654), 321 (BL#29655), 324 (BL#25182), 350 (BL#28176), 352, 355 (BL#55038), 356 (BL#28178), 357 (BL#25184), 358 (BL#25185), 359 (BL#28179), 361 (BL#28180), 365 (BL#25445), 373 (BL#28184), 379 (BL#25189), 380 (BL#25190), 385 (BL#28191), 386 (BL#28192), 390 (BL#55021), 391 (BL#25191), 392 (BL#28194), 399 (BL#25192), 405 (BL#29656), 409 (BL#28278), 426 (BL#28196), 437 (BL#25194), 439 (BL#29658), 443 (BL#28199), 486 (BL#25195), 502 (BL#28204), 513 (BL#29659), 517 (BL#25197), 530 (BL#29660), 531 (BL#28207), 555 (BL#25198), 563 (BL#28211), 584 (BL#28212). Additional strains obtained from the BDSC are: *Tk-GAL4* [ref. [49]] (BL#51974), *UAS-CsChrimson* [ref. [52]] (BL#55136). The low and high aggression strains used in this manuscript are a laboratory stock of Canton-S flies and a high aggression strain derived from a Canton-S strain that was selected for hyper aggression for 54 generations[29,53] (H.A. D., unpublished data). After approximately 100 generations of relaxation, a third chromosome derivative of this high aggression selection line was generated through chromosome isolation with cantonized balancer lines described previously[24]. The third chromosome high aggression strain was then recombined with a low aggression Canton S strain third chromosome through random breeding for 15 generations after which individual males were isolated, balanced, and isogenized and subsequently tested for aggression. All flies were reared on cornmeal, molasses, sugar, yeast, and agar food, on a 16 h light/8 h dark cycle (unless otherwise specified), and at room temperature (22.5 ± 0.5 °C). All flies for behavioral experiments were grown in bottles with 20 pairs of parents that were flipped every 3–4 days.

### Behavioral assays

*Divider Assay setup.* The behavioral chambers were designed in standard freely available CAD software (www.tinkercad.com) and 3D printed from the online marketplace (3dhubs.com) in ABS material. All behavioral assays were carried out in 12 arena behavioral chambers unless otherwise specified. The dimensions of individual square arenas were 13 × 4.5 mm (*W × H*). The chambers were assembled as shown in Supplementary Video 1 on a clear food medium (clear corn syrup, sucrose, and agar) to provide the flies with a continuous food source. On the day of eclosion flies were anesthetized under $CO_2$, and gently loaded, one fly on either side of opaque dividers that split the square arenas in half, and once all flies were loaded the chamber was covered with a glass lid. The isolated flies were then aged in the arenas for 5 days, at which point the dividers were carefully removed allowing the flies to freely interact. All regular behavioral assays were carried out within the first 3 h of the daily light cycle. For assays in behavioral rhythmicity the dividers were removed at different times during the light/dark cycle. For these experiments, flies were reared on a 12 h light/12 h dark cycle. All behavioral experiments were carried out at ~22.5 °C and ~50% humidity.

In each Divider Assay, 12 pairs of 5-day-old isolated males of the same genotype were recorded for 20 min immediately after removal of the dividers that separated the flies in each arena. For assays of increased space the height of the arena was varied (3.5, 4.5, 7.5, 11 mm) with fixed area (13 mm × 13 mm), or the height was kept constant (4.5 mm) with increased area (30 mm × 30 mm (~5×), and 47 mm × 47 mm (~13×)). To examine the effect of age, we housed flies in the regular behavioral chambers as specified above, and removed the dividers after 1, 2, 3, or 5 days followed by 20 min of videotaping and lunge analysis. For 10- and 30-day-old flies, males were maintained in groups of 5 in fly vials for 7 and 27 days, respectively, and then isolated in behavioral chambers for 3 days because no significant differences in lunge numbers were observed between 3- and 5-day isolated flies. Fighting pairs were always drawn from different group housed vials.

*Arena and Colosseum assays.* Arena assays were performed with 7-day-old flies (4 days group housed, 3 days isolated in isolation vials) as previously described[39]. Briefly, males were collected on the day of eclosion, isolated in small tubes for 3 days, and tested in the Arena assay by aspirating pairs of flies into the Fluon-coated chamber set of a 2% agarose gel and videotaped. Lunges were manually counted. Colosseum assays were performed with 5-day-old socially naïve males which were isolated in vials (dimension 10 × 50 mm, with 0.5 mm regular food) as pupae. The fights were carried out in 12-well polystyrene plates, with centrally placed food cups[54]. For both Arena and Colosseum assays, pairs of male flies were gently aspirated into the respective behavioral chambers.

*Courtship assay.* Each behavior arena (described above) was loaded with one virgin female and one male on the day of eclosion and separated for 5 days in the arena over food. Twelve pairs of male–female pairs were simultaneously recorded for 10 min on the day of testing after removal of dividers.

*Optogenetic activation of Tk-GAL4 circuit.* Behavioral setup was as for the standard setup. Except that we added all-*trans*-retinal (ATR) to the food at a final concentration of 500 μM. Flies were lighted from below with standard strong white light. *Tk-GAL4 > UAS-CsChrimson* flies not treated with ATR showed a weaker

response. We commonly observe a weaker response in no ATR-treated flies expressing this Channelrhodopsin variant.

**Video capture and data extraction**. Video data were collected using Basler acA1920—155 μm and Basler acA 2000—165 μm NIR cameras (Graftek Imaging). Behavioral chambers assembled on clear food were gently placed on top of an LED light source for high contrast image captures. For recording in total darkness, infrared LEDs were used instead of a regular LED light source. Frames were captured using Pylon Viewer 5 software at 20 Hz, and converted to movies (.avi format) using in house MATLAB code. PylonViewer also has a movie recording mode in the recent update (PylonViewer version 5.1.0.12681 64-bit). Movies with multiple fly pairs were tracked simultaneously using FlyTracker v1.0.5 software[35], and tracked movies were analyzed in JAABA (Janelia Automated Animal Behavior Analysis) software[36], both run in MATLAB. The lunge classifier (Lunge.jab) was designed in JAABA and used to mark lunging behaviors (single frames) displayed by individual flies. Annotated frames were postprocessed in JAABA with the internal post-processing filter set at 0.04, a value that provided the best signal to noise ratio for lunge classification. We designed an additional post-processing filter in MATLAB using JAABA postprocessed files in combination with tracking data to eliminate remaining misclassified lunges based on inter-fly distance of two or more fly body-lengths. Raw lunge number data per fly were exported as excel (v16.16.13,.csv) files. Courtship index, latency to court/mate, and mating frequency were manually calculated from courtship assay videos.

**Statistics and reproducibility**. Correlation analysis was performed using Pearson's $r$, followed by $R^2$ calculation and $p$ value determination. Because aggression and courtship data are not normally distributed, we used nonparametric tests for all statistical comparisons. Pairwise comparisons were carried out using Wilcoxon rank-sum tests. For comparisons of three groups or more, we used Kruskal–Wallis ANOVA followed by post hoc tests to determine statistically significantly different groups. Multiple comparisons with controls were carried out using Steel's method (with appropriate control groups), and multiple comparisons without a priori control groups were carried out with Dunn's test (reported significant $p$ values showing Bonferroni correction). All statistical analyses were performed in JMP pro v13 (except Supplementary Fig. 1). All nonparametric data are plotted as boxplots showing the median, first and third quartiles as boxes, with whiskers representing the 5 and 95% intervals. All individual data points are visualized on the boxplots as dots. Each dot represents an independent pair of flies. To better visualize the performance difference between our classifier and CADABRA lunge number analysis, we generated estimation plots of the effect size with confidence intervals determined by 5000 bootstrap samples of the raw data. We used the online resource https://www.estimationstats.com[41] with default parameters in this web application generating Gardner–Altman plots with two sided $p$ values calculated with nonparametric Mann–Whitney $U$ tests. All replicates are indicated in each figure. All pairs were independent and represent biological replicates.

**Reporting summary**. Further information on research design is available in the Nature Research Reporting Summary linked to this article.

## Data availability
Raw data are included in the source data file (Supplementary Data 1) and video files are available from the authors upon request. 3D-print files for the divider assay chamber are provided in Supplementary Data 2 and 3.

## Code availability
All code is available at the following links:
https://drive.google.com/file/d/1BvhUA41misBW1iHCDfACmJvXwgCoYjSU/view
https://github.com/budhaChowdhury/dividerAssay_codes

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

## Acknowledgements

Stocks obtained from the Bloomington Drosophila Stock Center (NIH P40OD018537) were used in this study. We thank the Bloomington stock center for strains from the DGRP collection. We thank Dan Park and Craig Kim for assistance with the manual lunge scoring, and Drs. Mansi Karkhanis, Christophe Herman, Hoang Nguyen, and Fabrizio Gabbiani for discussions and critical comments on the manuscript. This work was funded by grants from the National Institutes of Health (RO1 GM109938 and RO1 MH107474 to H.A.D.).

## Author contributions

J.P.G. and H.A.D. designed the Divider Assay with minor modifications from B.C. B.C. created the JAABA "lunge" classifier. B.C. and H.A.D. designed the experiments. J.P.G. created the high aggression strain and did the preliminary experiments on aging, time of day effect, and DGRP strains (together with H.A.D.). B.C., M.W., and H.A.D. carried out the experiments presented here. B.C. and H.A.D. analyzed the data, and wrote the manuscript with comments from J.P.G.

## Competing interests

The authors declare no competing interests.
