## [Peer Review File · Communications Biology]

Reviewers' comments:

Reviewer #1 (Remarks to the Author):

This is an interesting manuscript from the Dierick laboratory that introduces a novel method for high throughput studies of aggression in *Drosophila*. The authors have developed an experimental procedure that they called Divider Assay and allows for analyzing aggressive behavior in flies over long periods of time and in the absence of environmental light. Although there are several versions of high throughput chambers from different laboratories and also some automatic methods for tracking behavioral patterns, including lunging, this method (the combination of chamber design and tracking system) is potentially more useful since it allows for the quantification of behavior in the dark, and their classifier could be in theory adapted for the quantification of other behavioral patterns. In addition, when comparing the accuracy of their method with others available, the Divider Assay seems to be more accurate. This claim seems convincing and well supported. However, the extent to which this method is actually faster and easier to implement is not clear and remains to be corroborated by different users. While some technical aspects seem like a clear improvement from alternative options, like the fact that there is no need to isolate the flies previous to the experiment, there are multiple steps involved in the Divider Assay and several different software required for the analysis.

The authors demonstrate the utility of their assay through a variety of experimental conditions. While some of these observations are quite interesting, like the existence of a daily rhythm in the intensity of fighting, others are minor or well known in the field (like the fact that aggression decreases with increased space). In addition, the observation that assay duration affects aggression is not novel since it has already been published and is well established (Yurkovic et al. 2006, not cited in this manuscript). The manuscript would benefit significantly from focusing on the most interesting biological observations, the behavioral rhythm, and sending at least some of the less relevant observations to supplementary materials (for example the fact that their hyperaggressive flies continue to be hyperaggressive when they are old, which is not very surprising). Overall, I think that this manuscript has the potential for contributing with a very useful tool to the field. However, I have several concerns.

Major issues:

1- The authors describe in detail the advantages of their assay over existing methods: 'Our new pipeline is faster, more accurate, and easier to implement than existing methods' (line 67). While the accuracy aspect is well supported by the data (Fig 1 c-e), it is not at all clear that this method is actually faster or easier to implement than the alternatives. Needless to say, any automatic tracking will be faster than manual scoring, but other automatic tracking methods have fewer steps and do not involve as many different programs for analysis (for one obvious case see Fig 1b) and therefore probably have fewer requirements for the computers involved. For this manuscript to be truly helpful to the community, the authors should openly compare also the disadvantages of each method, including their own.

For example:

- In terms of the Divider Assay chambers, the fact that the food is all over the arena and there isn't a focal source makes it unlikely that a key behavioral parameter such as establishment of dominance can be quantified. Also, the fact that latency to lunge in the Colosseum Assay is shorter (likely as pointed out due to the presence of the food cup) despite the well-known fact that flies are less aggressive in larger chambers says something important about the use of environments in which aggressive behavior is exhibited in less stressful conditions than an extremely reduced space such as the chambers described in this manuscript. It is clear that small chamber dimensions are a

requirement for any high throughput system, but this doesn't mean that this assay is better overall. Different assays are suitable for tackling different biological questions and this should be explicitly addressed.

- In terms of the experimental design, it is not clear whether this setup would allow to perform experiments using common genetic tools such as shibirets and TrpA1 which are key for the study of behavior in *Drosophila* and require sharp increases in environmental temperature. This is possible using both the Anderson lab's and the Kravitz lab's arenas, among others. If their method can't be used for experiments involving such commonly used tools, it should be mentioned.

2- I agree with the authors that the MAS score and the method used for analyzing interactions in the related experiments mentioned (groups of 4-8 males analyzed for a very short amount of time) are not an a very effective way to measure aggression. However, the claim that 'the MAS score does not capture the main behavioral feature associated with aggression in *Drosophila*' (lines 244-245) is based on the fact that they can't reproduce those findings in their assay, which seems arbitrary. To support the notion that the phenotypes observed in the Divider Assay are reliable, the authors should at the very least test flies with increased levels of aggression reported by other laboratories, like the activation of tachykinin expressing neurons (Anderson Lab) or the feminization of cholinergic neurons (Kravitz lab) and see similar results.

3- The time of day effect is the most potentially interesting biological data shown in the manuscript but not entirely convincing in its current form. The effects observed could be merely due to the fact that control fights fight less in the absence of light. The notion that there is a rhythm in aggression would be more convincing -and interesting- if it persisted in the absence of environmental cues (DD or 'free running'). All the papers cited in this manuscript that have explored rhythms in social behaviors, particularly courtship & mating, have focused on DD and not LD experiments precisely for this reason. Since the Divider Assay allows to quantify aggressive behavior in the dark, it should be fairly easy to determine if the rhythm in aggression persists in DD.

4- Finally, the daily rhythm experiments should be compared with locomotor activity data to correlate levels of aggression with levels of activity at each of the chosen timepoints. The daily lunging profiles reported for the CS and hyperaggressive lines are quite different (Fig 6 a and b). Is this also the case for their locomotor activity profiles? Flies are the most active around the lights-off transition, the point at which the authors report the lowest levels of activity for their control, and show very low activity levels around ZT4, when the authors report the highest level of aggression. The abrupt drop in lunging seen at ZT12 resembles the phenotype observed by the Amrein lab for male courtship (Fujii et al, 2007) but contradicts what that same study showed for interactions in pairs of males. Also, and going back to the previous point (#3), based on the Amrein data there doesn't seem to be a clear rhythm in male-male pairs in DD. This makes it even more important to determine whether the rhythm in aggression persists under free running conditions or not.

Minor issues:

1- Some important references are missing, particularly work from Joel Levine's lab on courtship rhythms and Todd et al (2018) which shows a circadian rhythm in aggression in mice.

Reviewer #2 (Remarks to the Author):

Chowdury et al. presents a behavioral arena that permits the study of aggression in male *Drosophila*. Video recorded data are then analyzed using JAABA, a software published by another

group in 2012 (<https://www.nature.com/articles/nmeth.2281>). This arena - along with comparisons to other published approaches - as well as the analysis presented on the effects of genotype, age, and circadian time are valuable. I think this is the right venue for publication. However, there are a few major and minor changes that I would suggest before publication to make the work more appropriate and valuable for the community.

Major revisions-

1. Please fix the title to be accurate. Perhaps something like "An arena for high-throughput studies of *Drosophila* aggression" because:

- a. The novel aspect (arena) of this study is not "automated" but still operated by a human.
- b. The automated aspect (JAABA) has been published elsewhere by another group.

2. Please include Supplementary Videos to illustrate (1) what the raw behavioral data from the arena -including aggression lunges- actually look like, and (2) what the JAABA analyzed data looks like i.e., what do 'detected' lunges look like.

Reviewer #3 (Remarks to the Author):

1. General comments

The authors present a newly developed behavioural assay to facilitate the observation and analysis of *Drosophila* male-to-male aggression behaviour. This assay allows for 24h observation and can be managed in the dark. It further can be used as a behavioural assay for courtship, which is another asset that could be described in greater detail.

Although the assay eliminates all visual contact between the flies, aggression is mainly mediated by pheromonal cues. Furthermore, previous studies showed an influence of the auditory system on aggression behaviour (Versteven et al, 2017; Corthals, et al 2017). Is it guaranteed that the individual chambers are soundproofed so the aggression songs and other sounds or even olfactory cues are not interfering with the aggression behaviour of other flies?

2.1 Abstract

l. 40-41: "a dramatic drop at night" seems to be overselling the point. Since the circadian rhythm of *Drosophila* is well-known and the authors later reference a similar finding for courtship behaviour this finding is more expected than surprising.

2.2 Introduction

The introduction gives a background on different behavioural screens previously developed to analyse *Drosophila* aggression. The authors explain in detail why this new assay is an improvement compared to other established methods.

It however lacks a detailed description of *Drosophila* aggression behaviour and how it is mediated. It also fails to comprehensively explain why it is of interest to analyse aggression behaviour.

l. 63 - 64: *Drosophila* aggression consists of more behaviours than the mentioned lunging and boxing behaviour. A more detailed description would be beneficial for the introduction.

2.3 Results

The authors show that the newly developed Divider Assay performs equally well as the existing methods.

l. 78-82: This paragraph might be more fitting within the methods section as it mainly describes the different steps an aggression assay requires.

l. 82-84: This should be moved to the introduction, with an added explanation of the different aggression behaviours male *Drosophila* show.

l. 96-97: Maybe add

Wang, Liming, et al. "A common genetic target for environmental and heritable influences on aggressiveness in *Drosophila*." *Proceedings of the National Academy of Sciences* 105.15 (2008): 5657-5663.

as a reference.

l. 97-98: Please elaborate why a social isolation period of 5 days was chosen. In the literature the time seems to vary from 24 hrs to up to 15 days. Does longer isolation have a stronger effect?

l. 137 – 139: Is there data available with a low aggression *Drosophila* strain. As suggested by Baier et al, 2002 dopamine increase seems to lower aggression behaviour.

l. 165-166: This raises the question if time in isolation affects aggression levels.

l. 211 This title seems not really fitting, as aggression behaviour does consist of more behaviours than just lunges. As is described in l. 244-245 "MAS does not capture the main behavioural feature associated with aggression in *Drosophila*", so this might be a more accurate title for this section.

l. 267 Are 24h on a 12/12 light dark cycle enough to reset the circadian rhythm after being reared on 16/8 cycle?

2.4 Discussion

l. 301-311 and l. 337-342: these paragraphs feel redundant.

2.5 Methods

l. 347, 350: lines lack BL ID, are those not available?

l. 363: specify humidity levels

l. 402 wavelength of infrared LED? *Drosophila* photoreceptors can absorb up to 650 nm

3 Figures

Example for lunging behaviour?

Response to the Reviewers COMMSBIO-19-1848

Reviewers' comments:

We would like to thank all the reviewers for their overall positive assessment and constructive criticisms to improve our manuscript. We believe we have done that despite the challenging times of social distancing and reduced access to the lab during the last 5 months of Covid19. To address the reviewer critiques we have performed new experiments that resulted in 3 additional figures (1 main figure and 2 supplementary figures) and 1 supplementary movie to better illustrate our real-time behavior and the classifier-based analysis of lunging, i.e. the comparison between real-time and analyzed behavioral data (requested by Reviewer #2).

Below we have addressed each of the reviewer's comments separately and indicated the changes made to the manuscript. All changes in the manuscript are highlighted in the revised text.

Reviewer #1 (Remarks to the Author):

This is an interesting manuscript from the Dierick laboratory that introduces a novel method for high throughput studies of aggression in *Drosophila*. The authors have developed an experimental procedure that they called Divider Assay and allows for analyzing aggressive behavior in flies over long periods of time and in the absence of environmental light. Although there are several versions of high throughput chambers from different laboratories and also some automatic methods for tracking behavioral patterns, including lunging, this method (the combination of chamber design and tracking system) is potentially more useful since it allows for the quantification of behavior in the dark, and their classifier could be in theory adapted for the quantification of other behavioral patterns. In addition, when comparing the accuracy of their method with others available, the Divider Assay seems to be more accurate. This claim seems convincing and well supported. However, the extent to which this method is actually faster and easier to implement is not clear and remains to be corroborated by different users. While some technical aspects seem like a clear improvement from alternative options, like the fact that there is no need to isolate the flies previous to the experiment, there are multiple steps involved in the Divider Assay and several different software required for the analysis.

We thank the reviewer for the overall positive assessment and for recognizing the novelty of our work. Whether the chamber is truly better/faster will really only become clear as more users use it and we will therefore need to publish it. We believe that the evidence in the paper demonstrates our contention that it is indeed all the things we say it is, as we will elaborate below. We will also address the issue of "software complexity" below under our response to major comment #1.

The authors demonstrate the utility of their assay through a variety of experimental conditions. While some of these observations are quite interesting, like the existence of a daily rhythm in the intensity of fighting, others are minor or

well known in the field (like the fact that aggression decreases with increased space). In addition, the observation that assay duration affects aggression is not novel since it has already been published and is well established (Yurkovic et al. 2006, not cited in this manuscript).

Because part of our paper is comparative, our intent in addressing assay duration was focused on several methods that are published that use short assay times (Edwards et al., 2006; Zhou et al., 2008; Koganezawa et al., 2016). Our work suggests that this is not wise and we added this analysis in particular to contrast it to the behavioral assay from the McKay lab, which we further compare in the next figure (Fig. 5). We believe that two factors contribute to the poor performance of their assay: the very short assay time, and setting a low threshold for what an aggressive interaction is. (We elaborate on this point extensively in response to Reviewer #3 below as well.)

We respectfully disagree that the effect of assay duration was clearly demonstrated in Yurkovic et al and therefore did not cite the paper. That paper is focused on how quickly dominance is established between any pair of flies (which is perhaps why some people have chosen to use very short assay times) and that the recognition of the opponent is specific and long lasting (and thus remembered). In fact a corollary of our argument here is that early behavioral interactions are not particularly predictive of later behavioral interactions.

The manuscript would benefit significantly from focusing on the most interesting biological observations, the behavioral rhythm, and sending at least some of the less relevant observations to supplementary materials (for example the fact that their hyperaggressive flies continue to be hyperaggressive when they are old, which is not very surprising).

We respectfully disagree and think that the development of our new chamber and accompanying classifier are the most interesting parts of the paper, as this will make high throughput screening for aggression directly a possibility for the first time ever. Nevertheless, the rhythm data are interesting as well and we have done more work, described below, to address several important suggestions brought up by the reviewer.

As to the second point, work by others has shown that most behaviors dramatically decline with age (Simon et al., 2006; Iliadi and Boulianne, 2010). Therefore seeing significant aggression in 30-d old flies is at least somewhat surprising although we agree this is not a major point, but still worth making in the main paper. In addition, old age aggression has never been reported before and it is thus also a novel finding worthy of being in the main figures. To satisfy the reviewer, we have done new experiments to expand on the circadian/light switch phenotype in an extra figure later in the manuscript (Fig. 7), although we would like to emphasize our paper is primarily methodological and not a circadian biology paper. We have tried to demonstrate the usefulness of our new method by looking at many parameters and several different existing systems. Ultimately it will be the implementation of our assay in different labs that will truly establish its broader usefulness. It is of course our hope and expectation that will be the case.

Overall, I think that this manuscript has the potential for contributing with a very useful tool to the field. However, I have several concerns.

Major issues:

1- The authors describe in detail the advantages of their assay over existing methods: 'Our new pipeline is faster, more accurate, and easier to implement than existing methods' (line 67). While the accuracy aspect is well supported by the data (Fig 1 c-e), it is not at all clear that this method is actually faster or easier to implement than the alternatives. Needless to say, any automatic tracking will be faster than manual scoring, but other automatic tracking methods have fewer steps and do not involve as many different programs for analysis (for one obvious case see Fig 1b) and therefore probably have fewer requirements for the computers involved. For this manuscript to be truly helpful to the community, the authors should openly compare also the disadvantages of each method, including their own.

This is an important point to address. We do believe that we have provided significant evidence to show that the assay is "faster, more accurate, and easier to implement." In our hands, assembly of the chamber and loading flies (steps 1-3) takes approximately 2-3 min (as we mention in Fig. 1a, "<5 min" as a realistic number for an inexperienced experimenter). Collecting, isolating, making isolation vials, and aspirating flies into the assay, as is done typically in other assays, takes approximately 15-20 min for an experienced behaviorist for the equivalent number of flies, a difference of 5- to 10-fold. The tracking of the flies with FlyTracker and the analysis of the tracked video file with our classifier in the JAABA platform are also faster than tracking and analysis in CADABRA. While there are more steps listed in Fig. 1b for our classifier based tracking and analysis, it actually does not take more time to execute these steps because they depend on a 2-step program. Both software packages run in MatLab as separate code files, as is the case for CADABRA, i.e. tracking and analysis are separate steps with separate code initiated one after the other. However, tracking with QTRACKS and analysis in CADABRA take much longer than in our system. In addition, CADABRA runs in an old version of Matlab that cannot be upgraded, which is not ideal as upgrades are often automatic (so care has to be taken to avoid upgrading or the system stops working altogether). More importantly, in our hands the tracking step with QTRACKS fails approximately 30% of the time and thus needs to be repeated until it successfully tracks (sometimes it fails repeatedly and tracking cannot be done period). Tracking one behavioral chamber with QTRACKS, in our experience, takes overnight, while tracking with FlyTracker can be done in a few hours (as short as 2 hours with a high performing processor). Thus while the software seems more complicated, it is not and it is in fact faster in the JAABA platform.

It is clear from our comparisons in the manuscript that our JAABA classifier is far more accurate than CADABRA, particularly for flies that lunge at high level (Fig. 1c-d). We have not compared our classifier to other JAABA classifiers for

aggression (developed by others) because these classifiers are not publicly available. Moreover, it is important when developing a classifier to optimize it for the context of the chamber that is used for behavioral analysis. So a different classifier may not be optimal for a different assay context. This is important because our chamber differs significantly from other chambers: our arenas are square and they are thin. For this reason, we have trained/developed a high performing classifier to accompany our new chamber and have optimized it for our assay, demonstrating high sensitivity and high specificity for lunges, the key behavioral element in aggression. Finally, we have collectively developed and/or used at least 6 different assays for aggression analysis (Dierick and Greenspan, 2006; Dierick, 2007; Chen et al., 2002; Trannoy, 2015; our new Divider assay), and we believe this assay is the easiest to work with based on our subjective experiences. We are also developing a step-by-step protocol to make the assay easy to learn for others so they can implement it in their labs with all the caveats and trouble shooting steps so that even a novice can analyze aggression without much training. This paper is in preparation and we plan to submit it as soon as our current paper is accepted. We did not include this in the current manuscript because it would make the paper much longer and our current paper is not a protocol but a method paper with comparisons and demonstrated applications.

The final point is also to some extent subjective, but we believe this is the best system to perform high throughput screening for fly aggression and we don't see any real disadvantages for screening. There is currently no existing platform compatible with high throughput aggression screening capability. Despite the contention by Dankert et al (2009) that CADABRA "should enable large scale screens for genes and neural circuits controlling courtship and aggression" no such screens have been published in more than 10 years since the publication of their paper using CADABRA. So far, my group is the only one that has published a chemical mutagenesis screen for genes affecting aggression and we used a proxy phenotype to do so, wing damage as we found that group housed aggressive flies damage each other's wings (Davis et al., 2018). Our EMS mutant wing damage screen of 1500 X mutant strains took 3 months, plus 2 months to analyze the wing damage hits for aggression. The only circuit screen published to date by Hoopfer et al, 2015 used the JAABA platform and the chamber developed in Dankert et al (2009), which is a direct derivative from the chamber developed in Dierick 2007 (so there was only minimal innovation there in the chamber). The Hoopfer screen took several years to screen the 1900 lines (Eric Hoopfer, personal communication at CSH Meeting 2015). With the appropriate infrastructure, we believe that the Divider Assay should perform better than our previous wing damage screen. We cannot find any disadvantages in the Divider Assay for high throughput screening, and thus have listed none. However, we concede that if one would like to analyze other parameters than lunging, other classifiers would have to be developed. If one would want to analyze aggression in different contexts, with food, in larger environments, in the presence of females, in larger groups or in a more naturalistic context, new systems or adaptations would have to be developed to do so. We believe that publishing our Divider Assay will likely stimulate such developments.

For example:

- In terms of the Divider Assay chambers, the fact that the food is all over the arena and there isn't a focal source makes it unlikely that a key behavioral parameter such as establishment of dominance can be quantified. Also, the fact that latency to lunge in the Colosseum Assay is shorter (likely as pointed out due to the presence of the food cup) despite the well-known fact that flies are less aggressive in larger chambers says something important about the use of environments in which aggressive behavior is exhibited in less stressful conditions than an extremely reduced space such as the chambers described in this manuscript. It is clear that small chamber dimensions are a requirement for any high throughput system, but this doesn't mean that this assay is better overall. Different assays are suitable for tackling different biological questions and this should be explicitly addressed.

We respectfully disagree on two points. Dominance is typically established between male flies that fight, but it does not constitute a key factor in quantifying the behavior. Dominance is based on identifying the fly that does all the kicking. We find that in low aggression strains, flies that do fight have stable dominance, but in high aggression strains, dominance tends to flip more often because the opponent does not subjugate easily. In other words, who is dominant is not that important in a quantitative sense, but how much flies kick each other is a better reflection of their aggression levels.

To precisely assess aggression in a strain, quantifying lunges is undeniably the most important parameter. This does not mean that anything else is unimportant, but from the perspective of someone who wants to identify mutants in large-scale screens, lunges should be the first parameter to analyze. I would like to remind the reviewers that we know virtually nothing about the underlying molecular components of aggression in flies, and even in other organisms our knowledge of mechanisms is very limited. I do not believe that this paper is the right venue to discuss all these nuances and this would be better included in a review for example.

- In terms of the experimental design, it is not clear whether this setup would allow to perform experiments using common genetic tools such as shibirets and TrpA1 which are key for the study of behavior in *Drosophila* and require sharp increases in environmental temperature. This is possible using both the Anderson lab's and the Kravitz lab's arenas, among others. If their method can't be used for experiments involving such commonly used tools, it should be mentioned.

We have not used temperature sensitive effectors because our experience is that flies are quite sensitive to temperature shifts, which can strongly affect aggression levels and certainly makes the analysis much noisier. In fact some of the published work using these effectors for aggression is very questionable in my opinion. We prefer to use other effectors such as channelrhodopsin (see also our comments on the next point) and we demonstrate that our method is

compatible with its use (new Fig. S3). We see no reason why our chamber would not be compatible to those other effector methods with all the caveats mentioned above, but leave it to others to test this since we avoid using temperature effectors to measure aggression.

2- I agree with the authors that the MAS score and the method used for analyzing interactions in the related experiments mentioned (groups of 4-8 males analyzed for a very short amount of time) are not an a very effective way to measure aggression. However, the claim that 'the MAS score does not capture the main behavioral feature associated with aggression in *Drosophila*' (lines 244-245) is based on the fact that they can't reproduce those findings in their assay, which seems arbitrary. To support the notion that the phenotypes observed in the Divider Assay are reliable, the authors should at the very least test flies with increased levels of aggression reported by other laboratories, like the activation of tachykinin expressing neurons (Anderson Lab) or the feminization of cholinergic neurons (Kravitz lab) and see similar results.

It is surprising that the reviewer on the one hand agrees that MAS score poorly reflects aggression, but on the other hand finds it arbitrary [!] that we do not find any correlation with their data in our assay. I would argue that our results are the first evidence that MAS is a poor measure for aggression. Our assay clearly does capture aggression reliably as demonstrated everywhere in the paper starting in figure 1, where we show that we have very few false positives and very few false negatives. However, to demonstrate to the reviewer that our assay does not just work on our strains, we have now included analysis of the Tk-GAL4 driver using UAS-CsChrimson as the effector. These data show that we can capture aggression in that strain compared to controls. We have included these results in a new supplementary Figure S3.

3- The time of day effect is the most potentially interesting biological data shown in the manuscript but not entirely convincing in its current form. The effects observed could be merely due to the fact that control fights fight less in the absence of light. The notion that there is a rhythm in aggression would be more convincing -and interesting- if it persisted in the absence of environmental cues (DD or 'free running'). All the papers cited in this manuscript that have explored rhythms in social behaviors, particularly courtship & mating, have focused on DD and not LD experiments precisely for this reason. Since the Divider Assay allows to quantify aggressive behavior in the dark, it should be fairly easy to determine if the rhythm in aggression persists in DD.

This is an important point raised by the reviewer because light/vision is indeed an important factor influencing aggression (first shown by Hoyer et al., 2008) and could potentially explain/contribute to the LD profile. We have performed two sets of experiments to address this issue presented in our new Fig. 7. In panel a, we have analyzed the hyper-aggressive flies under DD conditions as requested by the reviewer. These results show that the circadian variation that is seen in LD is still seen under 'free running' DD conditions, although the lowest point of aggression occurs earlier at CT8 instead of ZT12 in

LD. Because the levels of aggression are already much lower in the hyper-aggressive strain (in the dark than in light), we did not believe running these experiments on wild-type flies would allow us to reliably detect differences for such low levels.

We also tested the effect of turning lights off and back on after different amounts of dark exposure time and thus also at different times of the day, presented in panel b. These results show that 'lights off' results in an immediate steep decrease in aggression even in the early morning when aggression is normally high. As flies stay in the dark for longer periods of time, their aggression level goes up slightly although not significantly, so there is some adaptation but it does not reach significance levels. All flies in this experiment were run in the dark or in light after the same 'dark treatment' and aggression in all cases fully rebounded immediately with no significant differences between any of the dark time points and between any of the light time points. This suggests that the darkness effect in LD is largely due to the light switch, but not entirely because although the effect is strongest in the first light switch dark time point, dark adaptation itself does not significantly increase aggression. In contrast different DD time points do show significant differences suggesting that some of the effect is indeed due to circadian modulation. What the cause of the light switch is, mechanistically, is not clear. It could be simply that flies cannot see their opponent and therefore are less able to attack, but is also possible that the 'drive' to fight is altered acutely through circuit activity changes in the relevant circuits. If so, it is unclear at this time what those circuits are and how they are affected. Future experiments will be needed to sort this out. We now discuss this in the revised discussion of the manuscript.

We went a step beyond what the reviewer asked and crossed a *per*⁰¹ mutant into our high aggression mutant background (a non-trivial and time consuming experiment) and found it to have very low aggression levels. We tested them only during the day-time points because we have not been allowed to enter the lab after hours since the shutdown and Phase I reopening due to Covid19. The double mutant shows very low levels of aggression (medians of ~5 lunges) such that variations are not interpretable. Whether the flies have low aggression because they are sick or whether the low aggression is mechanistically meaningful is unclear at the moment. While potentially interesting, examining the molecular components and the neural circuits of the clock is clearly beyond the scope of this manuscript and will have to be more thoroughly analyzed in the future.

4- Finally, the daily rhythm experiments should be compared with locomotor activity data to correlate levels of aggression with levels of activity at each of the chosen timepoints. The daily lunging profiles reported for the CS and hyperaggressive lines are quite different (Fig 6 a and b). Is this also the case for their locomotor activity profiles? Flies are the most active around the lights-off transition, the point at which the authors report the lowest levels of activity for their control, and show very low activity levels around ZT4, when the authors report the highest level of aggression. The abrupt drop in lunging seen at ZT12

resembles the phenotype observed by the Amrein lab for male courtship (Fujii et al, 2007) but contradicts what that same study showed for interactions in pairs of males. Also, and going back to the previous point (#3), based on the Amrein data there doesn't seem to be a clear rhythm in male-male pairs in DD. This makes it even more important to determine whether the rhythm in aggression persists under free running conditions or not.

We consider this possibility very unlikely for several reasons. First, the strains are derived from strains originally selected for changes in aggression (Dierick and Greenspan, 2006). In that manuscript we profiled sleep (based on locomotor data of flies in the TriKinetics DAM system) in these strains under LD and DD conditions and found no differences between the strains in the averages or in the profiles. There is no correlation between locomotion of single flies and the amount of aggression they display in pairs or groups.

Second, the derivative hyper-aggressive strain is a more recently re-derived version of the selected line in which all chromosomes were cantonized and the third chromosome was recombined with Canton S for 17 generations before re-deriving the recombinant chromosome that is responsible for the high aggression phenotype. Again, when we did these experiments, we analyzed the flies for sleep and found no differences.

Finally, our courtship and mating analysis of these strains under LD conditions also show no remarkable differences between the two strains suggesting that locomotion is not a key parameter difference between the strains. We would also like to emphasize once more that our manuscript is not a circadian paper. The title is focused on the methodology as a high throughput pipeline and we demonstrate that in several ways including with the day/night profile analysis.

Minor issues:

1- Some important references are missing, particularly work from Joel Levine's lab on courtship rhythms and Todd et al (2018) which shows a circadian rhythm in aggression in mice.

We thank the reviewer for these references and we have included them in our revised version.

Reviewer #2 (Remarks to the Author):

Chowdury et al. presents a behavioral arena that permits the study of aggression in male *Drosophila*. Video recorded data are then analyzed using JAABA, a software published by another group in 2012

(<https://www.nature.com/articles/nmeth.2281>). This arena - along with comparisons to other published approaches - as well as the analysis presented on the effects of genotype, age, and circadian time are valuable. I think this is the right venue for publication. However, there are a few major and minor changes that I would suggest before publication to make the work more appropriate and valuable for the community.

We thank the reviewer for their overall positive assessment.

Major revisions-

1. Please fix the title to be accurate. Perhaps something like "An arena for high-throughput studies of *Drosophila* aggression" because:

- a. The novel aspect (arena) of this study is not "automated" but still operated by a human.
- b. The automated aspect (JAABA) has been published elsewhere by another group.

We agree with the reviewer's first point that our chamber is not automated because a person still has to load it and the word "automated" should not be in the title. I would like to point out that an automated system without human intervention would likely be very expensive and therefore difficult to implement in any fly lab because of the cost and complicated controls. Our assay is cheap and easy to implement in any lab.

As to point b, we don't think that the only innovation here is the chamber. It is the key part, but the classifier is also important. We did indeed use existing software packages and acknowledge that clearly throughout the paper. However, the original JAABA paper does not have a classifier for aggression (JAABA itself does not score aggression) and other aggression JAABA classifiers are not publicly available for us to test in this context so our classifier is a critical component of the innovative chamber. As we mention above, a classifier should always be optimized to the context of the assay although we show our classifier works well in deeper and larger chambers and we have found that it also performs well in circular chambers (data not shown). We don't think investigators should use circular chambers however, because flies run in circles in circular chambers and this reduces their tendency to fight. In addition, circular chamber set-ups for aggression require treatment of the chamber to prevent flies from running on the walls. This treatment introduces another chemical parameter that can influence behavior, can be variably efficiently applied from chamber to chamber, and also requires time to apply, which makes the assay slower to execute.

For all these reasons, we suggest as our new title "A high throughput pipeline for aggression analysis in *Drosophila*."

2. Please include Supplementary Videos to illustrate (1) what the raw behavioral data from the arena -including aggression lunges- actually look like, and (2) what the JAABA analyzed data looks like i.e., what do 'detected' lunges look like.

We have now included an additional supplementary video file addressing both points from the reviewer (Supplementary Video 2). We show raw lunge and boxing behavior in section a, followed by slow motion video of the tracked flies in b, lunge classification/identification for fly 1 in c, and lunge classification for fly 2 in d. This demonstrates the classifier lunge detection in real time even when a fly is engaged in reciprocal lunging during a boxing bout.

Reviewer #3 (Remarks to the Author):

1. General comments

The authors present a newly developed behavioural assay to facilitate the observation and analysis of *Drosophila* male-to-male aggression behaviour. This assay allows for 24h observation and can be managed in the dark. It further can be used as a behavioural assay for courtship, which is another asset that could be described in greater detail.

Although the assay eliminates all visual contact between the flies, aggression is mainly mediated by pheromonal cues. Furthermore, previous studies showed an influence of the auditory system on aggression behaviour (Versteven et al, 2017; Corthals, et al 2017). Is it guaranteed that the individual chambers are soundproofed so the aggression songs and other sounds or even olfactory cues are not interfering with the aggression behaviour of other flies?

We can use the Divider Assay for courtship behavior but is not yet optimized for this use. We are currently working on a modified design that works more optimally for courtship. This will also require a completely new classifier training to capture they key courtship parameters, which will then have to be tested and rigorously compared to a manual observer.

We concede that we cannot rule out pheromonal and/or auditory “communication” between the separated flies. However, our evidence shows that isolation is nevertheless effective. To further illustrate this point, we have done an additional experiment (new Fig. S2). We tested flies that are 5 days old after 5d isolation and after 2d isolation to evaluate whether the difference in aggression with 2d old flies is due primarily to a difference in age or a difference in isolation. These experiments suggest that the difference in aggression (although not statistically significant between any of the groups) is due to isolation rather than age as the medians are nearly identical between 2d old and isolated versus 5d old and 2d isolated flies. This underscores that isolation is effective regardless of whether it blocks pheromonal and/or auditory cues. I would like to make an additional point here. The work on the effect of auditory cues by Versteven et al, mentioned by the reviewer, is based on an assay very similar to the MacKay assay, which we demonstrate in our paper has no correlation with lunge number analysis, the most important component in aggression quantification (also see below). It is therefore in my opinion at present questionable whether auditory cues indeed affect aggression. If that is the case, auditory separation may not be important.

2.1 Abstract

l. 40-41: “a dramatic drop at night” seems to be overselling the point. Since the circadian rhythm of *Drosophila* is well-known and the authors later reference a similar finding for courtship behaviour this finding is more expected than surprising.

Our intent was not to call the effect dramatic in the artistic sense of the word but as a qualifier of strength of the effect. We have changed the text to “a drastic drop at night.”

2.2 Introduction

The introduction gives a background on different behavioural screens previously developed to analyse *Drosophila* aggression. The authors explain in detail why this new assay is an improvement compared to other established methods. It however lacks a detailed description of *Drosophila* aggression behaviour and how it is mediated. It also fails to comprehensively explain why it is of interest to analyse aggression behaviour.

We have added a sentence in the first paragraph of the introduction to elaborate on the behavioral elements of aggression in flies.

Our paper is methodological in nature and justifying why one would want to study aggression is not really the point. We present a method for people to use that is in our view an easier method than any that currently exists and we believe we provide strong evidence for this argument. What the motivation for an investigator to study aggression is and whether it is worth it to do so, is not our role to discuss. That said, aggression is a truly complex behavior that seems to be regulated by specific circuitry and mechanisms and yet we still know very little about how it works. As we have argued in a previous review article (Thomas et al, 2015), aggression seems to be pervasive in nature, which means that it evolved many times or has ancient roots. Some evidence suggests indeed that at least some of its roots are ancient and *Drosophila* is perhaps the best organism to rapidly dissect the underlying molecular components through genetic screens. Unfortunately, no great screening methods currently exist. Our method addresses this gap. I believe this is the essence of the argument in the introduction and I think it needs no expansion.

I. 63 – 64: *Drosophila* aggression consists of more behaviours than the mentioned lunging and boxing behaviour. A more detailed description would be beneficial for the introduction.

As mentioned above, we have added a short explanation on the behavioral elements in the introduction with appropriate references in case the reader wants to delve in deeper. Below we give a more detailed answer to the importance of different behavioral elements in the quantification of aggression.

2.3 Results

The authors show that the newly developed Divider Assay performs equally well as the existing methods.

I. 78-82: This paragraph might be more fitting within the methods section as it mainly describes the different steps an aggression assay requires.

We respectfully disagree. This paragraph sets up the reasoning behind our motivation to develop the new assay in the first place. If one does not know what the problem is, how can one fix it? We think it is necessary to provide that argument up front.

I. 82-84: This should be moved to the introduction, with an added explanation of the different aggression behaviours male *Drosophila* show.

We again respectfully disagree. This sentence drives home the argument to focus on lunging for further analysis. Even with the elaboration in the introduction, it is worth emphasizing the point here because it is the reasoning behind our classifier development.

I. 96-97: Maybe add Wang, Liming, et al. "A common genetic target for environmental and heritable influences on aggressiveness in *Drosophila*." Proceedings of the National Academy of Sciences 105.15 (2008): 5657-5663. as a reference.

We have added Wang et al. as a citation here in the revised text.

I. 97-98: Please elaborate why a social isolation period of 5 days was chosen. In the literature the time seems to vary from 24 hrs to up to 15 days. Does longer isolation have a stronger effect?

There are indeed many different isolation duration variations in the literature. We picked 5 days because at 5d both control and hyper-aggressive seem to have the highest medians. Because the whole point of our assay is to make it easier and faster, we load the flies on the day of eclosion, which makes them by default 5d isolated when they are 5d old (although the collecting and loading step do not have to be done simultaneously). In other words, 5d is long enough and not too long. We did the aging experiment to show how the behavior changes with age and gives investigators a baseline to start from should they want to use a different time frame.

I. 137 – 139: Is there data available with a low aggression *Drosophila* strain. As suggested by Baier et al, 2002 dopamine increase seems to lower aggression behaviour.

We consider our control wild-type strain a low aggression strain. Lunge numbers are quite low in Canton S. They typically lunge less than 20 times in 20 min. In our lab, we consider strains with such low lunge numbers non aggressive. For screening purposes, we are only interested once strains go significantly above that level. We also believe that identifying genetic factors that decrease aggression are not particularly interesting, or more problematically, even reliable. There are many more ways to lose aggression non-specifically than to increase aggression. For example, if a fly has no legs, one would never be able to measure its aggression levels even if it would lunge at extremely high levels were it to have legs. So interventions that decrease aggression are only reliably aggression regulating/modulating if one can show that the reverse treatment increases aggression. While one would lose some interesting dampening factors this way that work one directionally in their modulatory effect, it is clear that reducing aggression is more difficult to assess as a "real" effect and many more experiments are required to better understand such an effect. In addition, finding very low lunge levels raises the problem of a floor effects. To address this, one could change the food in the chamber to increase the food value and the drive to fight. We plan to address this point in our protocol paper that we are currently developing.

As an aside, I do not believe that DA has any effect on aggression despite the literature arguing otherwise, at least in a global sense as shown in Baier et al. I have thoroughly examined this long ago by altering DA levels (and measuring the changes in DA in the fly head) alone or in combination with other biogenic amines in low and high aggression strain backgrounds and I have never found a noticeable statistically significant effect of DA. I think DA alterations have little to no effect on aggression. Perhaps individual DA neurons do affect aggression, but I have not seen any convincing evidence of this to date in any of the published work.

I. 165-166: This raises the question if time in isolation affects aggression levels.

As mentioned above, we did a new experiment to examine this. As shown in new Fig. S2 and discussed above, we tested flies that are 5 days old after 5d isolation and after 2d isolation to evaluate whether the difference in aggression with 2d old flies is due primarily to a difference in age or a difference in isolation. Our findings suggest that longer isolation does increase aggression.

I. 211 This title seems not really fitting, as aggression behaviour does consist of more behaviours than just lunges. As is described in I. 244-245 "MAS does not capture the main behavioural feature associated with aggression in *Drosophila*", so this might be a more accurate title for this section.

I agree with the reviewer that MAS does not capture lunges, but I also think it does not capture other behavioral elements that can convincingly be categorized as aggression, justifying the title of this results section. Let me elaborate. Historically, aggression in *Drosophila* was first defined by 3 unambiguous parameters: wing threat, lunging, and boxing. Lunging has also been called charging, and boxing has also been called tussling. A charge is a lunge that starts with an accelerated approach where the fly uses its wings in a rowing motion to propel itself forward to gain speed (Jacobs, 1960; clearly shown in Dierick, 2007). A lunge is therefore a better definition of the parameter because lunging does not require the charge part or the accelerated approach and can happen when the flies are almost stationary as in a boxing bout where flies move with variable speed and distance. In addition, accelerating towards a fly is not sufficient to unambiguously identify aggression as males often accelerate towards females when they perform courtship (although it is currently unknown whether there is a difference in maximum speed between these two accelerations). Boxing is almost always reciprocal lunging, but can also involve flies grabbing each other and sometimes looks like wrestling because flies seem to do whatever it takes to get the upper hand, which is likely true for any fighting animal. Holding is variant endpoint of a lunge where the fly instead of bouncing back, grabs the wings or another body part of the opponent. Therefore, quantification of lunging captures the vast majority of the behavior that can be unambiguously defined as aggressive, and which can either be one-directional or reciprocal (i.e. executed by one fly or by both flies). These behavioral categories are tight and easily recognizable and teachable to a behavioral experimenter. Interestingly, the first two papers that clearly described the aggressive behaviors

in male flies described the exact same above mentioned 3 unambiguous behavioral elements based on their descriptions, but named them somewhat differently (Jacobs, 1960; Dow and von Schilcher, 1975). I believe that Dow and von Schilcher likely were unaware of the Jacobs paper (1960) when they wrote their Nature paper (1975) since they do not cite Jacobs, and yet they described the same behaviors underscoring how solid these classifications are for an experienced observer of this behavior. Later work by Hoffmann (1987, 1988 etc) again used very similar behavioral classifications as in the work from 1960 and 1975. Chen et al (2002) introduced low level fencing and high level fencing to the repertoire and Nilsen et al (2004) later further expanded/renamed this to low-posture and high-posture fencing. If you examine their Markov chain analysis, there is no connection between low posture fencing and lunging and the transition of high posture fencing to lunging is also rare (these transitions are not even shown in Chen et al., 2002). They also see rare connections between lunging and boxing because they analyzed wild-type flies that have low aggression levels, and rarely box or reciprocate, and rarely transition from lunging to boxing. In high aggression strains boxing and lunging are common high probability transitions and they represent more than 90% of the aggressive behaviors. This is not because flies have nowhere to go in the limited space of enclosed arenas because these behavioral transitions are extremely frequent in population cages where flies have all the space in the world. In fact, in population cages, boxing almost always results from lunging by one fly that is resisted by the other fly, which then suddenly turns around to escalate the event into reciprocal lunging (boxing). In other words, in a population setting boxing is not typically a transition from high posture fencing slowly escalating to reciprocal lunging, but from resisted lunging. These events often last for minutes, so the extreme escalated behavior is not the result of small spaces, quite the opposite, it appears to be the result of a drive to control the food patch. Even in low aggression strains, lunging represents the vast majority of the behavior although it is almost always one-directional in such low aggression strains. If one eliminates what I would call non-aggressive behavioral elements from the Markov chain analysis in Nilsen et al (in Chen et al lunging is missing from the analysis altogether), analyzed on these low aggression flies, almost all that is left is lunging, some boxing (at low levels), holding and wing threat and high posture fencing. And, as explained above, all of these fall in the category of lunging or boxing, which itself is mostly made up of reciprocal lunging. None of these elements appear to be part of MAS quantification. A link to "low" and "high aggression" strain analysis from the MacKay lab has been posted by a former student of theirs, which I have pasted here below. I urge the reviewer to analyze these videos and find anything that looks like unambiguous aggression. Also compare these clips to the video added to our supplementary materials. It appears that what MacKay et al quantify is leg touching between flies. Anyone who has observed pairs or groups of flies for any significant amount of time knows that one of the major forms of fly communication is touch. In fact, it is how flies instantly identify the sex of other flies and how they communicate in ways we do not yet understand. The vast majority of these touches never escalate to

aggressive encounters of any kind. In other words, what the MAS assay likely scores is touch, which is almost always unrelated to aggression, which explains the absence of any correlation between MAS and lunge numbers. Think about the following: despite the fact that the MAS quantifications differ by more than 20-fold across the strains we tested, we find not one strain with lunge levels that even reach the levels seen in a wild-type low aggression strain. Even if one were to accept that some of the touches represent aggression, it is clear that the vast majority do not progress to aggression, which means that MAS is not a reliable measure for any aggression because it cannot distinguish between those that do and don't progress to aggression. Evaluate the evidence yourself:

High Aggression: <https://www.youtube.com/watch?v=D3vzTTEIGoA>

Low Aggression: <https://www.youtube.com/watch?v=yM43aI4HZIq>

I. 267 Are 24h on a 12/12 light dark cycle enough to reset the circadian rhythm after being reared on 16/8 cycle?

As we mentioned in the text, for these experiments, we maintained flies in a 12/12 light dark cycle regimen. All flies were tested on day 5 and the following night.

2.4 Discussion

I. 301-311 and I. 337-342: these paragraphs feel redundant.

We agree that it is a bit redundant but it is the final message we want to reiterate for our reader.

2.5 Methods

I. 347, 350: lines lack BL ID, are those not available?

Yes, for two of the lines we could not find a BL ID, we think BDSC must have lost these lines sometime after we obtained them.

I. 363: specify humidity levels

Humidity levels are maintained at 50%.

For your information, while humidity can affect circadian entrainment, there is no correlation between humidity level variation and aggression as far as our analyses show.

I. 402 wavelength of infrared LED? Drosophila photoreceptors can absorb up to 650 nm

The infrared LED has a wavelength of 850 nm.

https://www.amazon.com/gp/product/B01DM9AT9C/ref=ppx_yo_dt_b_asin_title_o01_s00?ie=UTF8&psc=1

3 Figures

Example for lunging behaviour?

As mentioned above to address the 2nd point of Reviewer #2, we have added a supplementary video file illustrating the behavior as well as classifier performance.

REVIEWERS' COMMENTS:

Reviewer #1 (Remarks to the Author):

The authors have addressed my experimental concerns and I recommend the manuscript for publication.

Reviewer #2 (Remarks to the Author):

The authors addressed my concerns.